# Why Spectral Normalization Stabilizes GANs: Analysis and Improvements

**Zinan Lin**
Carnegie Mellon University
Pittsburgh, PA 15213
zinanl@andrew.cmu.edu

**Vyas Sekar**
Carnegie Mellon University
Pittsburgh, PA 15213
vsekar@andrew.cmu.edu

**Giulia Fanti**
Carnegie Mellon University
Pittsburgh, PA 15213
gfanti@andrew.cmu.edu

## Abstract

Spectral normalization (SN) [30] is a widely-used technique for improving the stability and sample quality of Generative Adversarial Networks (GANs). However, current understanding of SN's efficacy is limited. In this work, we show that SN controls two important failure modes of GAN training: exploding and vanishing gradients. Our proofs illustrate a (perhaps unintentional) connection with the successful LeCun initialization [25]. This connection helps to explain why the most popular implementation of SN for GANs [30] requires no hyper-parameter tuning, whereas stricter implementations of SN [15, 12] have poor empirical performance out-of-the-box. Unlike LeCun initialization which only controls gradient vanishing at the beginning of training, SN preserves this property throughout training. Building on this theoretical understanding, we propose a new spectral normalization technique: Bidirectional Scaled Spectral Normalization (BSSN), which incorporates insights from later improvements to LeCun initialization: Xavier initialization [13] and Kaiming initialization [17]. Theoretically, we show that BSSN gives better gradient control than SN. Empirically, we demonstrate that it outperforms SN in sample quality and training stability on several benchmark datasets.

## 1 Introduction

Generative adversarial networks (GANs) are state-of-the-art deep generative models, perhaps best known for their ability to produce high-resolution, photorealistic images [14]. The objective of GANs is to produce random samples from a target data distribution, given only access to an initial set of training samples. This is achieved by learning two functions: a generator $G$, which maps random input noise to a generated sample, and a discriminator $D$, which tries to classify input samples as either real (i.e., from the training dataset) or fake (i.e., produced by the generator). In practice, these functions are implemented by deep neural networks (DNNs), and the competing generator and discriminator are trained in an alternating process known as *adversarial training*. Theoretically, given enough data and model capacity, GANs converge to the true underlying data distribution [14].

Although GANs have been very successful in improving the sample quality of data-driven generative models [22, 8], their adversarial training also contributes to instability. That is, small hyper-parameter changes and even randomness in the optimization can cause training to fail. Many approaches have been proposed for improving the stability of GANs, including different architectures [38, 22, 8], loss functions [2, 3, 16, 50], and various types of regularizations/normalizations [30, 9, 41]. One of the most successful proposals to date is called *spectral normalization* (SN) [30, 15, 12]. SN forces each layer of the discriminator to have unit spectral norm during training. This has the effect of controlling the Lipschitz constant of the discriminator, which is empirically observed to improve the stability of GAN training [30]. Despite the successful applications of SN [8, 28, 53, 21, 52, 31, 27], to date, it remains unclear precisely why this specific normalization is so effective.

35th Conference on Neural Information Processing Systems (NeurIPS 2021).

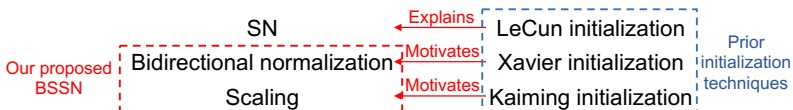

Figure 1: The interesting connections we find between spectral normalizations and prior initialization techniques: (1) LeCun initialization [25] can help explain why SN avoids vanishing gradients; (2) Motivated by newer initialization techniques [13, 17], we propose BSSN to further improve SN.

In this paper, we show that SN controls two important failure modes of GAN training: exploding gradients and vanishing gradients. These problems are well-known to cause instability in GANs [3, 8], leading either to bad local minima or stalled training prior to convergence. We make three primary contributions:

*(1) Analysis of why SN avoids exploding gradients (§ 3).* Poorly-chosen architectures and hyper-parameters, as well as randomness during training, can amplify the effects of large gradients on training instability, ultimately leading to generalization error in the learned discriminator. We theoretically prove that SN upper bounds gradients during GAN training, mitigating these effects.

*(2) Analysis of why SN avoids vanishing gradients (§ 4).* Small gradients during training are known to cause GANs (and other DNNs) to converge to bad models [25, 3]. The well-known LeCun initialization, first proposed over two decades ago, mitigates this effect by carefully choosing the variance of the initial weights [25]. We prove theoretically that SN controls the variance of weights in a way that closely parallels LeCun initialization. Whereas LeCun initialization only controls the gradient vanishing problem at the beginning of training, we show empirically that SN preserves this property throughout training. Our analysis also explains why a strict implementation of SN [12] has poor out-of-the-box performance on GANs and requires additional tuning to avoid the vanishing gradient problem, whereas the implementation of SN in [30] requires no tuning.

*(3) Improving SN with the above theoretical insights (§ 5).* Given this new understanding of the connections between SN and LeCun initialization, we propose Bidirectional Scaled Spectral Normalization (BSSN), a simple modification of SN that combines two key insights (Fig. 1): (a) It introduces a novel bidirectional spectral normalization inspired by *Xavier initialization*, which improved on LeCun initialization by controlling not only the variances of internal outputs, but also the variance of backpropagated gradients [13]. We theoretically prove that BSSN mimics Xavier initialization to give better gradient control than SN. (b) BSSN introduces a new scaling of weights inspired by *Kaiming initialization*, a newer initialization technique that has better performance in practice [17]. We show that BSSN achieve better sample quality and training stability than SN on several benchmark datasets, including CIFAR10, STL10, CelebA, and ImageNet.

Note that our goal in this work is not to propose the best normalization or regularization technique for training GANs; there has been substantial work in this area already with promising results [34, 48, 20, 11]. Our goal is instead to understand *why* SN has been so effective, and evaluate whether these insights can inform more effective alternatives. Our theoretical and empirical results suggest that gradient control may play a significant role in this story (though they need not be the only factor explaining SN's success).

## 2 Background and Preliminaries

The instability of GANs is believed to be predominantly caused by poor discriminator learning [2, 40]. We therefore focus in this work on the discriminator, and the effects of SN on discriminator learning. We adopt the same model as [30]. Consider a discriminator with $L$ internal layers:

$$D_\theta(x) = a_L \circ l_{w_L} \circ a_{L-1} \circ l_{w_{L-1}} \circ \ldots \circ a_1 \circ l_{w_1}(x) \tag{1}$$

where $x$ denotes the input to the discriminator and $\theta = \{w_1, w_2, ..., w_L\}$ the weights; $a_i\, (i = 1, ..., L-1)$ is the activation function in the $i$-th layer, which is usually element-wise ReLU or leaky ReLU in GANs [14]. $a_L$ is the activation function for the last layer, which is sigmoid for the vanilla GAN [14] and identity for WGAN-GP [16]; $l_{w_i}$ is the linear transformation in $i$-th layer, which is usually fully-connected or a convolutional neural network [14, 38]. Like prior work on the theoretical analysis of (spectral) normalization [30, 12, 42, 20], we do not model bias terms.

**Lipschitz regularization and spectral normalization.** Prior work has shown that regularizing the Lipschitz constant of the discriminator $\|D_\theta\|_{\text{Lip}}$ improves the stability of GANs [3, 16, 50]. For example, WGAN-GP [16] adds a gradient penalty $(\|\nabla D_\theta(\tilde{x})\| - 1)^2$ to the loss function, where $\tilde{x} = \alpha x + (1 - \alpha)G(z)$ and $\alpha \sim \text{Uniform}(0, 1)$ to ensure that the Lipschitz constant of the discriminator is bounded by 1.

Spectral normalization (SN) takes a different approach. For fully connected layers (i.e., $l_{w_i}(x) = w_i x$), it regularizes the weights $w_i$ to ensure that spectral norm $\|w_i\|_{\text{sp}} = 1$ for all $i \in [1, L]$, where the spectral norm $\|w_i\|_{\text{sp}}$ is defined as the largest singular value of $w_i$. This bounds the Lipschitz constant of the discriminator since $\|D_\theta\|_{\text{Lip}} \leq \prod_{i=1}^{L} \|l_{w_i}\|_{\text{Lip}} \cdot \prod_{i=1}^{L} \|a_i\|_{\text{Lip}} \leq \prod_{i=1}^{L} \|w_i\|_{\text{sp}} \cdot \prod_{i=1}^{L} \|a_i\|_{\text{Lip}} \leq 1$, as $\|l_{w_i}\|_{\text{Lip}} \leq \|w_i\|_{\text{sp}}$ and $\|a_i\|_{\text{Lip}} \leq 1$ for networks with (leaky) ReLU as activation functions for the internal layers and identity/sigmoid as the activation function for the last layer [30]. Prior work has theoretically connected the generalization gap of neural networks to the product of the spectral norms of the layers [5, 32, 20, 48]. These insights led to multiple implementations of spectral normalization [12, 15, 51, 30, 20, 48], with the implementation of [30] achieving particular success on GANs. SN can be viewed as a special case of more general techniques for enhancing the stability of neural network training by controlling the entire spectrum of a network's input-output Jacobian [34], weight matrices [20], or learned embeddings [48].

In practice, spectral normalization [12, 30] is implemented by dividing the weight matrix $w_i$ by its spectral norm: $\frac{w_i}{u_i^T w_i v_i}$, where $u_i$ and $v_i$ are the left/right singular vectors of $w_i$ corresponding to its largest singular value. As observed by Gouk et al. [15], there are two approaches in the SN literature for instantiating the matrix $w_i$ for convolutional neural networks (CNNs). In a CNN, since convolution is a linear operation, convolutional layers can equivalently be written as a multiplication by an expanded weight matrix $\tilde{w}_i$ that is derived from the raw weights $w_i$. Hence in principle, spectral normalization should normalize each convolutional layer by $\|\tilde{w}_i\|_{\text{sp}}$ [15, 12]. We call this canonical normalization $\text{SN}_{\text{Conv}}$ as it controls the spectral norm of the convolution layer.

However, the spectral normalization that is known to outperform other regularization techniques and improves training stability for GANs [30], which we call $\text{SN}_{\text{w}}$, does not implement SN in a strict sense. Instead, it uses $\left\|w_i^{c_{out} \times (c_{in} k_w k_h)}\right\|_{\text{sp}}$; that is, it first reshapes the convolution kernel $w_i \in \mathbb{R}^{c_{out} c_{in} k_w k_h}$ into a matrix $\hat{w}_i$ of shape $c_{out} \times (c_{in} k_w k_h)$, and then normalizes with the spectral norm $\|\hat{w}_i\|_{\text{sp}}$, where $c_{in}$ is the number of input channels, $c_{out}$ is the number of output channels, $k_w$ is the kernel width, and $k_h$ is the kernel height. Miyato et al. showed that their implementation implicitly penalizes $w_i$ from being too sensitive in one specific direction [30]. However, this does not explain why $\text{SN}_{\text{w}}$ is more stable than other Lipschitz regularization techniques, and as observed in [15], it is unclear how $\text{SN}_{\text{w}}$ relates to $\text{SN}_{\text{Conv}}$. Despite this, $\text{SN}_{\text{w}}$ has empirically been immensely successful in stabilizing the training of GANs [8, 28, 53, 21, 52, 31, 27]. Even more puzzling, we show in § 4 that the canonical approach $\text{SN}_{\text{Conv}}$ has comparatively poor out-of-the-box performance when training GANs.

Hence, two questions arise: (1) Why is SN so successful at stabilizing the training of GANs? (2) Why is $\text{SN}_{\text{w}}$ proposed by [30] so much more effective than the canonical $\text{SN}_{\text{Conv}}$?

In this work, we show that both questions are related to two well-known phenomena: vanishing and exploding gradients. These terms describe a problem in which gradients either grow or shrink rapidly during training [6, 35, 36, 7], and they are known to be closely related to the instability of GANs [2, 8]. We provide an example to illustrate how vanishing or exploding gradients cause training instability in GANs in App. I.

## 3 Exploding Gradients

In this section, we show that spectral normalization prevents gradient explosion by bounding the gradients of the discriminator. Moreover, we show that the common choice to normalize all layers equally achieves the tightest upper bound for a restricted class of discriminators. We use $\theta \in \mathbb{R}^d$ to denote a vector containing all elements in $\{w_1, ..., w_L\}$. In the following analysis, we assume linear transformations are fully-connected layers $l_{w_i}(x) = w_i x$ as in [30], though the same analysis can be applied to convolutional layers. Following prior work on the theoretical analysis of (spectral) normalization [30, 12, 42], we assume no bias in the network (i.e., Eq. (1)) for simplicity.

To highlight the effects of the spectral norm of each layer on the gradient and simplify the exposition, we will compute gradients with respect to $w'_i = \frac{w_i}{u_i^T w_i v_i}$ in the following discussion. In reality, gradients are computed with respect to $w_i$; we defer this discussion to App. C, where we show the relevant extension.

**How SN controls exploding gradients.** The following proposition shows that under this simplifying assumption, spectral normalization controls the magnitudes of the gradients of the discriminator with respect to $\theta$. Notice that simply controlling the Lipschitz constant of the discriminator (e.g., as in WGAN [2]) does not imply this property; it instead ensures small (sub)gradients with respect to the input, $x$.

**Proposition 1** (Upper bound of gradient's Frobenius norm for spectral normalization). *If $\|w_i\|_{sp} \leq 1$ for all $i \in [1, L]$, then we have $\|\nabla_{w_t} D_\theta(x)\|_F \leq \|x\| \prod_{i=1}^{L} \|a_i\|_{Lip}$, and the norm of the overall gradient can be bounded by $\|\nabla_\theta D_\theta(x)\|_F \leq \sqrt{L}\, \|x\| \prod_{i=1}^{L} \|a_i\|_{Lip}$.*

*(Proof in App. A).* Note that under the assumption that internal activation functions are ReLU or leaky ReLU, if the activation function for the last layer is identity (e.g., for WGAN-GP [16]), the above bounds can be simplified to $\|\nabla_{w_t} D_\theta(x)\|_F \leq \|x\|$ and $\|\nabla_\theta D_\theta(x)\| \leq \sqrt{L}\, \|x\|$, and if the activation for the last layer is sigmoid (e.g., for vanilla GAN [14]), the above bounds become $\|\nabla_{w_t} D_\theta(x)\|_F \leq 0.25\, \|x\|$ and $\|\nabla_\theta D_\theta(x)\| \leq 0.25\sqrt{L}\, \|x\|$. A comparable bound can also be found to limit the norm of the Hessian, which we defer to App. D.

The bound in Prop. 1 has a significant effect in practice. Fig. 2 shows the norm of the gradient for each layer of a GAN trained on MNIST with and without spectral normalization. Without spectral normalization, some layers have extremely large gradients throughout training, which makes the overall gradient large. With spectral normalization, the gradients of all layers are upper bounded as shown in Prop. 1. We see similar results in other datasets and network architectures (App. J).

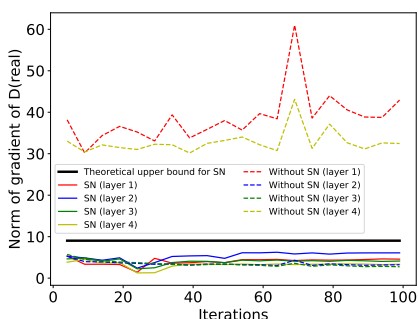

Figure 2: Gradient norms of each discriminator layer in MNIST.

**Optimal spectral norm allocation.** Common implementations of SN advocate setting the spectral norm of *each layer* to the same value [30, 12]. However, the following proposition states that we can set the spectral norms of different layers to different constants, without changing the network's behavior on the input samples, as long as the *product* of the spectral norm bounds is the same.

**Proposition 2.** *For any discriminator $D_\theta = a_L \circ l_{w_L} \circ a_{L-1} \circ l_{w_{L-1}} \circ \ldots \circ a_1 \circ l_{w_1}$ and $D'_\theta = a_L \circ l_{c_L \cdot w_L} \circ a_{L-1} \circ l_{c_{L-1} \cdot w_{L-1}} \circ \ldots \circ a_1 \circ l_{c_1 \cdot w_1}$ where the internal activation functions $\{a_i\}_{i=1}^{L-1}$ are ReLU or leaky ReLU, and positive constant scalars $c_1, ..., c_L$ satisfy that $\prod_{i=1}^{L} c_i = 1$, we have*

$$D_\theta(x) = D'_\theta(x) \;\forall x \quad and \quad \frac{\partial^n D_\theta(x)}{\partial x^n} = \frac{\partial^n D'_\theta(x)}{\partial x^n} \;\forall x, \forall n \in \mathbb{Z}^+ .$$

*(Proof in App. B).* Given this observation, it is natural to ask if there is any benefit to setting the spectral norms of each layer equal. It turns out that the answer is yes, under some assumptions that appear to approximately hold in practice. Let

$$\mathcal{D} \triangleq \left\{ D_\theta = a_L \circ l_{w_L} \circ \ldots \circ a_1 \circ l_{w_1} \;:\; \frac{\|\nabla_{w_i} D_\theta(x)\|_F}{\|\nabla_{w_j} D_\theta(x)\|_F} = \frac{\|w_j\|_{sp}}{\|w_i\|_{sp}}, a_i \in \{\text{ReLU, leaky ReLU}\} \;\forall i, j \in [1, L] \right\}. \quad (2)$$

This intuitively describes the set of all discriminators for which scaling up the weight of one layer proportionally increases the gradient norm of all other layers; the definition of this set is motivated by our upper bound on the gradient norm (App. A). The following theorem shows that when optimizing over set $\mathcal{D}$, choosing every layer to have the same spectral norm gives the smallest possible gradient norm, for a given set of parameters.

**Theorem 1.** *Consider a given set of discriminator parameters $\theta = \{w_1, ..., w_L\}$. For a vector $c = \{c_1, \ldots, c_L\}$, we denote $\theta_c \triangleq \{c_t w_t\}_{t=1}^{L}$. Let $\lambda_\theta = \prod_{i=1}^{L} \|w_i\|_{sp}^{1/L}$ denote the geometric mean*

*of the spectral norms of the weights. Then we have*

$$\left\{ \frac{\lambda_\theta}{\|w_1\|_{sp}}, \ldots, \frac{\lambda_\theta}{\|w_L\|_{sp}} \right\} = \arg \min_{c:\ D_{\theta_c} \in \mathcal{D},\ \prod_{i=1}^{L} c_i = 1,\ c_i \in \mathbb{R}^+} \|\nabla_{\theta_c} D_{\theta_c}(x)\|_F$$

*(Proof in App. E).* The key constraint in this theorem is that we optimize only over discriminators in set $\mathcal{D}$ in Eq. (2). To show that this constraint is realistic (i.e., SN GAN discriminator optimization tends to choose models in $\mathcal{D}$), we trained a spectrally-normalized GAN with four hidden layers on MNIST, computing the ratios of the gradient norms at each layer and the ratios of the spectral norms, as dictated by Eq. (2). We computed these ratios at different epochs during training, as well as for different randomly-selected rescalings of the spectral normalization vector $c$. Each point in Fig. 3 represents the results averaged over 64 real samples at a specific epoch of training for a given (random) $c$. Vertical series of points are from different epochs of the same run, therefore their ratio of spectral norms is the same. The fact that most of the points are near the diagonal line suggests that training naturally favors discriminators that are in or near $\mathcal{D}$; we confirm this intuition in other experimental settings in App. K. This observation, combined with Thm. 1, suggests that it is better to force the spectral norms of every layer to be equal. Hence, existing SN implementations [30, 12] chose the correct, uniform normalization across layers to upper bound discriminator's gradients.

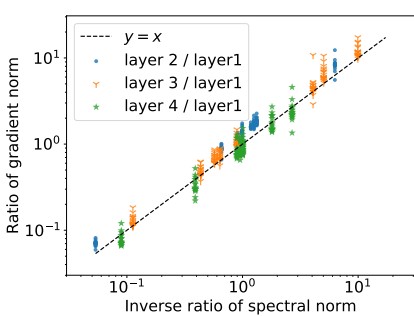

Figure 3: Ratio of gradient norm v.s. inverse ratio of spectral norm in MNIST.

**Implications on other normalization/regularization techniques.** Note that the analysis in this section can also be used to show the same results for other normalization/regularization techniques that control the spectral norm of weights, like weight normalization [41] and orthogonal regularization [9]. However, these techniques do not necessarily exhibit the more important properties proved in § 4 for SN, and have some known drawbacks (see more discussion in § 6).

## 4  Vanishing Gradients

An equally troublesome failure mode of GAN training is vanishing gradients [2]. Prior work has proposed new objective functions to mitigate this problem [2, 3, 16], but these approaches are not fully effective (Fig. 11). In this section, we show that SN also helps to control vanishing gradients.

**How SN controls vanishing gradients.** Gradients tend to vanish for two reasons. First, gradients vanish when the objective function *saturates* [25, 2], which is often associated with function parameters growing too large. Common loss functions (e.g., hinge loss) and activation functions (e.g., sigmoid, tanh) saturate for inputs of large magnitude. Large parameters tend to amplify the inputs to the activation functions and/or loss functions, causing saturation. Second, gradients vanish when function parameters (and hence, internal outputs) grow too small. This is because backpropagated gradients are scaled by the function parameters (App. A).

These insights motivated the LeCun initialization technique [25]. The key idea is that to prevent gradients from vanishing, we must ensure that the outputs of each neuron do not vanish or explode. If the inputs to a neural unit are uncorrelated random variables with variance 1, then to ensure that the unit's output also has variance (approximately) 1, the weight parameters should be zero-mean random variables with variance of $\frac{1}{n_i}$, where $n_i$ denote the fan-in (number of incoming connections) of layer $i$ [25]. Hence, LeCun initialization prevents gradient vanishing by controlling the variance of the individual parameters. In the following theorem, we show that SN enforces a similar condition.

**Theorem 2** (Parameter variance of SN)**.** *For a matrix $A \in \mathbb{R}^{m \times n}$ with i.i.d. entries $a_{ij}$ from a symmetric distribution with zero mean (e.g., zero-mean Gaussian or uniform), we have*

$$Var\left( \frac{a_{ij}}{\|A\|_{sp}} \right) \leq \frac{1}{\max\{m,n\}} \quad . \tag{3}$$

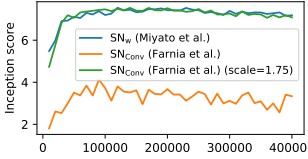

Figure 4: Inception score of different SN variants in CIFAR10.

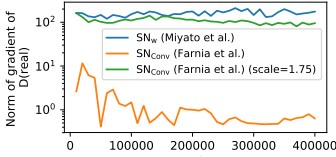

Figure 5: Gradient norms of different SN variants in CIFAR10.

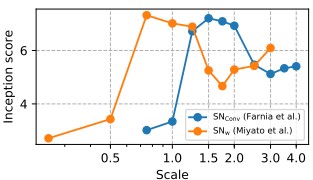

Figure 6: Inception score of scaled SN in CIFAR10.

*Furthermore, if $m, n \geq 2$ and $\max\{m, n\} \geq 3$, and $a_{ij}$ are from a zero-mean Gaussian, we have*

$$\frac{L}{\max\{m,n\} \log(\min\{m,n\})} \leq Var\left(\frac{a_{ij}}{\|A\|_{sp}}\right) \leq \frac{1}{\max\{m,n\}} \quad ,$$

*where $L$ is a constant which does not depend on $m, n$.*

*(Proof in App. F)*. Hence, spectral normalization forces zero-mean parameters to have a variance that scales inversely with $\max\{m, n\}$. The proof relies on a characterization of extreme values of random vectors drawn from the surface of a high-dimensional unit ball. Many fully-connected, feed-forward neural networks have a fixed width across hidden layers, so $\max\{m, n\}$ corresponds precisely to the fan-in of any neuron in a hidden layer, implying that SN has an effect like LeCun initialization.

**Why $SN_w$ works better than $SN_{Conv}$.** In a CNN, the interpretation of $\max\{m, n\}$ depends on how SN is implemented. Recall that the implementation $SN_w$ by [30] does not strictly implement SN, but a variant that normalizes by the spectral norm of $\hat{w}_i = w_i^{c_{out} \times (c_{in}k_wk_h)}$. In architectures like DCGAN [38], the larger dimension of $\hat{w}_i$ for *hidden layers* tends to be $c_{in}k_wk_h$, which is exactly the fan-in. This means that SN gets the right variance for hidden layers in CNN.

Perhaps surprisingly, we find empirically that the strict implementation $SN_{Conv}$ of [12] does *not* prevent gradient vanishing. Figs. 4 and 5 shows the gradients of $SN_{Conv}$ vanishing when trained on CIFAR10, leading to a comparatively poor inception score, whereas the gradients of $SN_w$ remain stable. To understand this phenomenon, recall that $SN_{Conv}$ normalizes by the spectral norm of an expanded matrix $\tilde{w}_i$ derived from $w_i$. Thm. 2 does not hold for $\tilde{w}_i$ since its entries are not i.i.d. (even at initialization); hence it cannot be used to explain this effect. However, Corollary 1 in [47] shows that $\|\hat{w}_i\|_{sp} \leq \|\tilde{w}_i\|_{sp} \leq \alpha \|\hat{w}_i\|_{sp}$, where $\alpha$ is a constant only depends on kernel size, input size, and stride size of the convolution operation. ([45] also deduced a special case of the second inequality.) This result has two implications:

(1) $\|\tilde{w}_i\|_{sp} \leq \alpha \|\hat{w}_i\|_{sp}$: Although $SN_w$ does not strictly normalize the matrix with the actual spectral norm of the layer, it does upper bound the spectral norm of the layer. Therefore, all our analysis in § 3 still applies for $SN_w$ by changing the spectral norm constant from 1 to $\alpha \|\hat{w}_i\|_{sp}$. This means that $SN_w$ can still prevent gradient explosion.

(2) $\|\hat{w}_i\|_{sp} \leq \|\tilde{w}_i\|_{sp}$: This implies that $SN_{Conv}$ normalizes by a factor that is at least as large as $SN_w$. In fact, we observe empirically that $\|\tilde{w}_i\|_{sp}$ is strictly larger than $\|\hat{w}_i\|_{sp}$ during training (App. L.3). This means that for the same $w_i$, a discriminator using $SN_{Conv}$ will have smaller outputs than the discriminator using $SN_w$. We hypothesize that the different scalings explain why $SN_{Conv}$ has vanishing gradients but $SN_w$ does not.

To confirm this hypothesis, for $SN_w$ and $SN_{Conv}$, we propose to multiply all the normalized weights by a scaling factor $s$, which is fixed throughout the training. Fig. 6 shows that $SN_{Conv}$ seems to be a shifted version of $SN_w$. $SN_{Conv}$ with $s = 1.75$ has similar inception score (Fig. 4) to $SN_w$, as well as similar gradients (Fig. 5) and parameter variances (App. L.4) throughout training. This, combined with Thm. 2, suggests that $SN_w$ inherently finds the correct scaling for the problem, whereas "proper" spectral normalization $SN_{Conv}$ requires additional hyper-parameter tuning.

**SN has good parameter variances throughout training.** Our theoretical analysis only applies at initialization, when the parameters are selected randomly. However, unlike LeCun initialization which only controls the variance at initialization, we find empirically that Eq. (3) for SN appears to hold *throughout training* (Fig. 7). As a comparison, if trained without SN, the variance increases and the gradient decreases, which makes sample quality bad (App. L.2). This explains why in practice GANs trained with SN are stable *throughout training*. Formally extending our theoretical analysis to apply throughout training requires more complicated techniques, which we defer to future work.

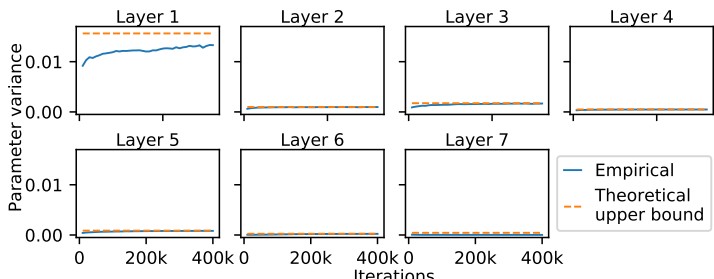

Figure 7: Parameter variances throughout training in CIFAR10. The blue lines show the parameter variances of different layers when SN is applied, and the original line shows our theoretical upper bound given in Eq. (3).

# 5 Extensions of Spectral Normalization

Given the above theoretical insights, we propose an extension of spectral normalization called Bidirectional Scaled Spectral Normalization (BSSN). It combines two key ideas: bidirectional normalization and weight scaling.

## 5.1 Bidirectional Normalization

Glorot and Bengio [13] built on the intuition of LeCun [25] to design an improved initialization, commonly called *Xavier initialization*. Their key observation was that to limit gradient vanishing (and explosion), it is not enough to control only feed-forward outputs; we should also control the variance of backpropagated gradients. Let $n_i, m_i$ denote the fan-in and fan-out of layer $i$. (In fully-connected layers, $n_i = m_{i-1} = $ the width of layer $i$.) Whereas LeCun chooses initial parameters with variance $\frac{1}{n_i}$, Glorot and Bengio choose them with variance $\frac{2}{n_i+m_i}$, a compromise between $\frac{1}{n_i}$ (to control output variance) and $\frac{1}{m_i}$ (to control variance of backpropagated gradients).

The first component of BSSN is Bidirectional Spectral Normalization (BSN), which applies a similar intuition to improve the spectral normalization of Miyato *et al.* [30]. For fully connected layers, BSN keeps the normalization the same as $SN_w$ [30]. For convolution layers, instead of normalizing by $\left\|w^{c_{out} \times (c_{in}k_w k_h)}\right\|_{sp}$, we normalize by $\sigma_w \triangleq \frac{\left\|w^{c_{out} \times (c_{in}k_w k_h)}\right\|_{sp} + \left\|w^{c_{in} \times (c_{out}k_w k_h)}\right\|_{sp}}{2}$, where $\left\|w^{c_{in} \times (c_{out}k_w k_h)}\right\|_{sp}$ is the spectral norm of the reshaped convolution kernel of dimension $c_{in} \times (c_{out}k_w k_h)$. For calculating these two spectral norms, we use the same power iteration method in [30]. The following theorem gives the theoretical explanation.

**Theorem 3** (Parameter variance of BSN). *For a convolutional kernel $w \in \mathbb{R}^{c_{out}c_{in}k_w k_h}$ with i.i.d. entries $w_{ij}$ from a symmetric distribution with zero mean (e.g. zero-mean Gaussian or uniform) where $k_w k_h \geq \max\left\{\frac{c_{out}}{c_{in}}, \frac{c_{in}}{c_{out}}\right\}$, and $\sigma_w$ defined as above, we have*

$$Var\left(\frac{w_{ij}}{\sigma_w}\right) \leq \frac{2}{c_{in}k_w k_h + c_{out}k_w k_h} \ .$$

*Furthermore, if $c_{in}, c_{out} \geq 2$ and $c_{in}k_w k_h, c_{out}k_w k_h \geq 3$, and $w_{ij}$ are from a zero-mean Gaussian distribution, there exists a constant $L$ that does not depend on $c_{in}, c_{out}, k_w, k_h$ such that*

$$\frac{L}{c_{in}k_w k_h \log(c_{out}) + c_{out}k_w k_h \log(c_{in})} \leq Var\left(\frac{w_{ij}}{\sigma_w}\right) \leq \frac{2}{c_{in}k_w k_h + c_{out}k_w k_h}.$$

*(Proof in App. G).* In convolution layers, $n_i = c_{in}k_w k_h$ and $m_i = c_{out}k_w k_h$. Therefore, BSN sets the variance of parameters to scale as $\frac{2}{n_i+m_i}$, as dictated by Xavier initialization. Moreover, BSN naturally inherits the benefits of SN discussed in § 4 (e.g., controlling variance throughout training).

## 5.2 Weight Scaling

The second component of BSSN is to multiply all the normalized weights by a constant scaling factor (i.e., as we did in Fig. 6). We call the combination of BSN and this weight scaling *Bidirectional Scaled Spectral Normalization* (BSSN). Note that scaling can also be applied independently to SN, which we call Scaled Spectral Normalization (SSN). The scaling is motivated by the following reasons.

(1) The analysis in LeCun and Xavier initialization assumes that the activation functions are linear, which is not true in practice. More recently, Kaiming initialization was proposed to include the effect of non-linear activations [17]. The result is that we should set the variance of parameters to be $2/(1 + a^2)$ times the ones in LeCun or Xavier initialization, where $a$ is the negative slope of leaky ReLU. This suggests the importance of a constant scaling.

(2) However, we found that the scaling constants proposed in LeCun/Kaiming initialization do not always perform well for GANs. Even more surprisingly, there are *multiple modes* of good scaling. Fig. 8 shows the sample quality of LeCun initialization with different scaling on the discriminator. We see that there are at least two good modes of scaling: one at around 0.2 and the other at around 1.2. This phenomenon cannot be explained by the analysis in LeCun/Kaiming initialization.

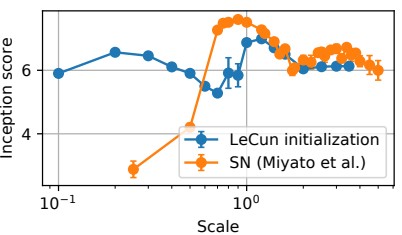

Recall that SN has similar properties as LeCun initialization (§ 4). Interestingly, we see that SSN also has two good modes of scaling (Fig. 8). Although the best scaling constants for LeCun initialization and SN are very different, there indeed exists an interesting mode correspondence in terms of parameter variances (App. M). We hypothesize that the shift of good scaling from Kaiming initialization we see here could result from adversarial training, and defer the theoretical analysis to future work. These results highlight the need for a separate scaling factor.

Figure 8: Inception score of SSN and scaled LeCun initialization in CIFAR10, with mean and standard error of the best score during training across multiple runs.

(3) The bounds in Thm. 2 and Thm. 3 only imply that in SN and BSN the *order* of parameter variance w.r.t. the network size is correct, but constant scaling is unknown.

## 5.3 Results

In this section we verify the effectiveness of BSSN with extensive experiments. The code for reproducing the results is at `https://github.com/fjxmlzn/BSN`.

[30] already compares SN with many other regularization techniques like WGAN-GP [16], batch normalization [19], layer normalization [4], weight normalization [41], and orthogonal regularization [9], and SN is shown to outperform them all. Therefore, we compare BSSN with SN. To isolate the effects of the two components proposed in BSSN, we also compare against bidirectional normalization without scaling (BSN) and scaling without bidirectional normalization (SSN).

We conduct experiments across different public (non-sensitive) datasets (from low-resolution to high-resolution) and network architectures (from standard CNN to ResNets). More specifically, we conducts experiments on CIFAR10, STL10, CelebA, and ImageNet (ILSVRC2012), following the same settings in [30]. All experimental details are attached in Apps. N to S. The results are in Table 1.

**BSN v.s. SN (showing the effect of bidirectional normalization § 5.1).** (1) By comparing BSN with SN in Table 1, we can see that BSN outperforms SN by a large margin in all metrics except in ILSVRC2012 (discussed later). (2) More importantly, the superiority of BSN is stable across hyper-parameters. In App. N, we vary the learning rates ($\alpha_g, \alpha_d$) and momentum parameters of generator and discriminator, and the number of discriminator updates per generator update ($n_{dis}$). We see that BSN consistently outperforms SN in most of the cases. (3) Moreover, BSN is more stable in the entire training process. We see that as training proceeds, the sample quality of SN often drops, whereas the sample quality of BSN appears to monotonically increase (Fig. 9, more in Apps. P to R).

| | CIFAR10 | | STL10 | | CelebA | ILSVRC2012 | |
|---|---|---|---|---|---|---|---|
| | IS $\uparrow$ | FID $\downarrow$ | IS $\uparrow$ | FID $\downarrow$ | FID $\downarrow$ | IS $\uparrow$ | FID $\downarrow$ |
| Real data | 11.26 | 9.70 | 26.70 | 10.17 | 4.44 | 197.37 | 15.62 |
| SN | $7.12 \pm 0.07$ | $31.43 \pm 0.90$ | $9.05 \pm 0.05$ | $44.35 \pm 0.54$ | $9.43 \pm 0.09$ | $12.84 \pm 0.33$ | $75.06 \pm 2.38$ |
| SSN | $7.38 \pm 0.06$ | $29.31 \pm 0.23$ | $\mathbf{9.28 \pm 0.03}$ | $43.52 \pm 0.26$ | $\mathbf{8.50 \pm 0.20}$ | $12.84 \pm 0.33$ | $73.21 \pm 1.92$ |
| BSN | $\mathbf{7.54 \pm 0.04}$ | $\mathbf{26.94 \pm 0.58}$ | $9.25 \pm 0.01$ | $42.98 \pm 0.54$ | $9.05 \pm 0.13$ | $1.77 \pm 0.13$ | $265.20 \pm 19.01$ |
| BSSN | $\mathbf{7.54 \pm 0.04}$ | $\mathbf{26.94 \pm 0.58}$ | $9.25 \pm 0.01$ | $\mathbf{42.90 \pm 0.17}$ | $9.05 \pm 0.13$ | $\mathbf{13.23 \pm 0.16}$ | $\mathbf{69.04 \pm 1.46}$ |

Table 1: Inception score (IS) and FID on CIFAR10, STL10, CelebA, and ILSVRC2012. The last three rows are proposed in this work, with BSSN representing our final proposal—a combination of BSN and SSN. Each experiment is conducted with 5 random seeds except that the last three rows on ILSVRC2012 is conducted with 3 random seeds. Mean and standard error across these random seeds are reported. We follow the common practice of excluding IS in CelebA as the inception network is pretrained on ImageNet, which is different from CelebA. Bold font marks best numbers in a column.

BSN generally outperforms SN in final sample quality (i.e., at the end of training), but also in *peak* sample quality. I.e., BSN stabilizes the training process, which is the purpose of SN (and BSN).

**SSN v.s. SN (showing the effect of scaling § 5.2).** Comparing SSN with SN in Table 1, we see that scaling consistently improves (or has the same metric) *in all cases*. This verifies our intuition in § 5.2 that the inherent scaling in SN is not optimal, and an extra constant scaling is needed for best results.

**BSSN v.s. BSN (showing the effect of scaling § 5.2).** By comparing BSSN with BSN in Table 1, we see that in some cases the optimal scale of BSN happens to be 1 (e.g., in CIFAR10), but in other cases, scaling is critical. For example, in ILSVRC2012, BSN without any scaling has the same gradient vanishing problem we observe for $SN_{Conv}$ [12] in § 4, which causes bad sample quality. BSSN successfully solves the gradient vanishing problem and achieves the best sample quality.

**Additional results.** Due to space constraints, we defer other supplementary results (e.g., generated images, training curves, more comparisons and analysis) to Apps. N to S.

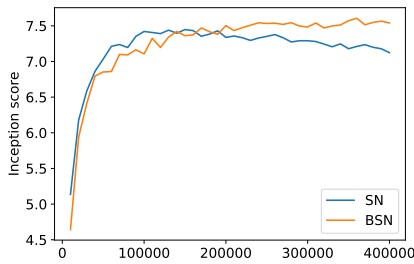

Figure 9: Inception score in CIFAR10. The results are averaged over 5 random seeds, with $\alpha_g = 0.0001$, $\alpha_d = 0.0001$, $n_{dis} = 1$.

**Summary.** In summary, both designs we proposed can effectively stabilize training and achieve better sample quality. Combining them together, BSSN achieves the best sample quality in most cases. This demonstrates the practical value of the theoretical insights in § 3 and 4.

## 6 Discussion

**Related work.** Related to our upper bound, [42] shows that batch normalization (BN) reduces the Hessian's scale along the gradient direction, making gradients more predictive. Given Prop. 1, we can apply this reasoning to explain why spectrally-normalized GANs are robust to different learning rates as shown in [30]. However, our insights regarding gradient vanishing are the more surprising result, and not discussed in [42]. It is unclear whether BN similarly controls vanishing gradients.

In parallel to this work, some other approaches have been proposed to improve SN. For example, [11] finds out that even with SN, the condition numbers of the weights can still be large, which causes instability. To solve the issue, they borrow the insights from linear algebra and propose preconditioning layers to improve the condition numbers and promote stability.

**Other reasons contributing to the stability of SN.** This paper presents one possible reason (i.e., SN avoids exploding and vanishing gradients), and shows such a correlation through theoretical and empirical analysis. However, there could exist many other parallel factors. For example, [30] points out that SN could speed up training by encouraging the weights to be updated along directions orthogonal to itself. Our results do not shed light on these orthogonal hypotheses.

**Implications on other normalization/regularization techniques.** As discussed in § 3, other normalization techniques like weight normalization [41] and orthogonal regularization [9] also control the maximum gradient norm, shown by a simple extension of our results. However, they perform worse in practice [30]. We hypothesize two reasons: (1) They may not have the more important properties proved in § 4. For example, the official implementation of orthogonal regularization on CNN kernels [1] gives a parameter variance of $\frac{1}{c_{in}k_w}$, which is larger than the one in $SN_w$; (2) They have known drawbacks like promoting less effective features [30]. Extending our analysis framework to explain these differences would be an interesting future work.

**Future directions.** Our results suggest that SN stabilizes GANs by controlling exploding and vanishing gradients in the discriminator. However, our analysis also applies to the training of any feed-forward neural network. This may explain why SN also helps train generators [53, 8] and neural networks more broadly [12, 15, 51]. We focus on GANs because SN seems to disproportionately benefit them [30]. Carefully understanding why is an interesting direction for future work.

Related to the weight initialization and training dynamics, recent work [37, 43] has shown that Gaussian weights or ReLU activations cannot achieve dynamical isometry (all singular values of the network Jacobian near 1), a desired property for training stability. Orthogonal weight initialization may be better at achieving the goal. We considered Gaussian weights and ReLU activations as they are the predominant implementations in GANs, but studying other networks may be useful too.

**Ethics.** Although our work focuses on understanding GAN fundamentals, some downstream applications of GANs (e.g., spreading misinformation) are ethically problematic. We believe there is still value to studying GANs, which can also enable positive applications like better synthetic data.

## Acknowledgements

This work was supported in part by faculty research awards from Google, JP Morgan Chase, and the Sloan Foundation, as well as gift grants from Cisco and Siemens AG. This research was sponsored in part by National Science Foundation Convergence Accelerator award 2040675 and the U.S. Army Combat Capabilities Development Command Army Research Laboratory and was accomplished under Cooperative Agreement Number W911NF-13-2-0045 (ARL Cyber Security CRA). The views and conclusions contained in this document are those of the authors and should not be interpreted as representing the official policies, either expressed or implied, of the Combat Capabilities Development Command Army Research Laboratory or the U.S. Government. The U.S. Government is authorized to reproduce and distribute reprints for Government purposes notwithstanding any copyright notation here on. Zinan Lin acknowledges the support of the Siemens FutureMakers Fellowship, the CMU Presidential Fellowship, and the Cylab Presidential Fellowship. This work used the Extreme Science and Engineering Discovery Environment (XSEDE) [46], which is supported by National Science Foundation grant number ACI-1548562. Specifically, it used the Bridges system [33], which is supported by NSF award number ACI-1445606, at the Pittsburgh Supercomputing Center (PSC).

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
