# A  Proof of Prop. 1

The proposition makes use of the following observation: For the discriminator defined in (1), the norm of gradient for $w_t$ is upper bounded by

$$\|\nabla_{w_t} D_\theta(x)\|_\mathrm{F} \leq \|x\| \cdot \prod_{i=1}^{L} \|a_i\|_\mathrm{Lip} \cdot \prod_{i=1}^{L} \|w_i\|_\mathrm{sp} \bigg/ \|w_t\|_\mathrm{sp} \quad \text{for } \forall t \in [1, L] \tag{4}$$

To prove this, for simplicity of notation, let $o_a^i = a_i \circ l_{w_i} \circ \ldots \circ a_1 \circ l_{w_1}$, and $o_l^i = l_{w_i} \circ a_{i-1} \circ \ldots \circ a_1 \circ l_{w_1}$.
It is straightforward to show that the norm of each internal output of discriminator is bounded by

$$\left\| o_a^t(x) \right\| \leq \|x\| \cdot \prod_{i=1}^{t} \|a_i\|_\mathrm{Lip} \cdot \prod_{i=1}^{t} \|w_i\|_\mathrm{sp} \tag{5}$$

and

$$\left\| o_l^t(x) \right\| \leq \|x\| \cdot \prod_{i=1}^{t-1} \|a_i\|_\mathrm{Lip} \cdot \prod_{i=1}^{t} \|w_i\|_\mathrm{sp} . \tag{6}$$

This holds because

$$\left\| o_a^t(x) \right\| = \left\| a_i \left( o_l^t(x) \right) \right\| \leq \|a_i\|_\mathrm{Lip} \cdot \left\| o_l^t(x) \right\|$$

and

$$\left\| o_l^t(x) \right\| = \left\| l_{w_i} \left( o_a^{t-1}(x) \right) \right\| \leq \|w_t\|_\mathrm{sp} \cdot \left\| o_a^{t-1}(x) \right\| ,$$

from which we can show the desired inequalities by induction.

Next, we observe that the norm of each internal gradient is bounded by

$$\left\| \nabla_{o_a^t(x)} D_\theta(x) \right\| \leq \prod_{i=t+1}^{L} \|a_i\|_\mathrm{Lip} \cdot \prod_{i=t+1}^{L} \|w_i\|_\mathrm{sp} \tag{7}$$

and

$$\left\| \nabla_{o_l^t(x)} D_\theta(x) \right\| \leq \prod_{i=t}^{L} \|a_i\|_\mathrm{Lip} \cdot \prod_{i=t+1}^{L} \|w_i\|_\mathrm{sp} . \tag{8}$$

This holds because

$$\left\| \nabla_{o_a^t(x)} D_\theta(x) \right\| = \left\| w_{t+1}^T \nabla_{o_l^{t+1}(x)} D_\theta(x) \right\| \leq \|w_{t+1}\|_\mathrm{sp} \left\| \nabla_{a_l^{t+1}(x)} D_\theta(x) \right\|$$

and

$$\left\| \nabla_{o_l^t(x)} D_\theta(x) \right\| = \left\| \left\langle \nabla_{o_a^t(x)} D_\theta(x), \left[ a_t'(x)|_{x=o_l^t(x)} \right] \right\rangle \right\| \leq \|a_t\|_\mathrm{Lip} \left\| \nabla_{o_a^t(x)} D_\theta(x) \right\| ,$$

from which we can show inequalities Eqs. (7) and (8) by induction.

Now we have that

$$\begin{aligned}
\|\nabla_{w_t} D_\theta(x)\|_\mathrm{F} &= \left\| \nabla_{o_l^t(x)} D_\theta(x) \cdot \left( o_a^{t-1}(x) \right)^\mathrm{T} \right\|_\mathrm{F} \\
&= \left\| \nabla_{o_l^t(x)} D_\theta(x) \right\| \cdot \left\| o_a^{t-1}(x) \right\| \\
&\leq \prod_{i=t}^{L} \|a_i\|_\mathrm{Lip} \cdot \prod_{i=t+1}^{L} \|w_i\|_\mathrm{sp} \cdot \|x\| \cdot \prod_{i=1}^{t-1} \|a_i\|_\mathrm{Lip} \cdot \prod_{i=1}^{t-1} \|w_i\|_\mathrm{sp} \\
&= \|x\| \cdot \prod_{i=1}^{L} \|a_i\|_\mathrm{Lip} \cdot \prod_{i=1}^{L} \|w_i\|_\mathrm{sp} \bigg/ \|w_t\|_\mathrm{sp}
\end{aligned}$$

where we use Eqs. (5) to (8) at the inequality. The upper bound of gradient's Frobenius norm for spectrally-normalized discriminators follows directly.

# B    Proof of Prop. 2

*Proof.* As $l_w(x)$ is a linear transformation, we have $l_{cw}(x) = c \cdot l_w(x)$, and $l_w(cx) = c \cdot l_w(x)$. Moreover, since ReLU and leaky ReLU is linear in $\mathbb{R}^+$ and $\mathbb{R}^-$ region, we have $a_i(cx) = c \cdot a_i(x)$. Therefore, we have

$$D'_\theta(x) = \left(a_L \circ l_{c_L \cdot w_L} \circ a_{L-1} \circ l_{c_{L-1} \cdot w_{L-1}} \circ \ldots \circ a_1 \circ l_{c_1 \cdot w_1}\right)(x)$$

$$= \prod_{i=1}^{L} c_i \cdot \left(a_L \circ l_{w_L} \circ a_{L-1} \circ l_{w_{L-1}} \circ \ldots \circ a_1 \circ l_{w_1}\right)(x)$$

$$= D_\theta(x)$$

$\square$

# C    Additional Analysis of Gradient

In § 3, we discuss the gradients with respect to $w'_i = \frac{w_i}{u_i^T w_i v_i}$, where $u_i, v_i$ are the singular vectors corresponding to the largest singular values. In this section we discuss the gradients with respect the actual parameter $w_i$. From Eq. (12) in [30] we know

$$\nabla_{w_t} D_\theta(x) = \frac{1}{\|w_t\|_{\mathrm{sp}}} \left(\nabla_{w'_t} D_\theta(x) - \left(\left(\nabla_{o_l^t(x)} D_\theta\left(x\right)\right)^T o_l^t\left(x\right)\right) \cdot u_t v_t^T\right)$$

From App. A, we know that $\left\|\nabla_{w'_t} D_\theta(x)\right\|_{\mathrm{F}}$, $\left\|\nabla_{o_l^t(x)} D_\theta\left(x\right)\right\|$, and $\left\|o_l^t\left(x\right)\right\|$ have upper bounds. Furthermore, $\left\|u_t v_t^T\right\|_{\mathrm{F}} = 1$. Therefore, $\left\|\nabla_{w'_t} D_\theta(x) - \left(\left(\nabla_{o_l^t(x)} D_\theta\left(x\right)\right)^T o_l^t\left(x\right)\right) \cdot u_t v_t^T\right\|_{\mathrm{F}}$ has an upper bound. From Theorem 1.1 in [44] we know that if $w_t$ is initialized with i.i.d random variables from uniform or Gaussian distribution, $\mathbb{E}\left(\|w_t\|_{\mathrm{sp}}\right)$ is lower bounded away from zero at initialization. So $\|\nabla_{w_t} D_\theta(x)\|_{\mathrm{F}}$ is upper bounded at initialization. Moreover, we observe empirically that $\|w_t\|_{\mathrm{sp}}$ is usually increasing during training. Therefore, $\|\nabla_{w_t} D_\theta(x)\|_{\mathrm{F}}$ is typically upper bounded during training as well.

# D    Analysis of Hessian

The following proposition states that spectral normalization also gives an upper bound on $\|H_{w_i}(D_\theta)(x)\|_{\mathrm{sp}}$ for networks with ReLU or leaky ReLU internal activations.

**Proposition 3** (Upper bound of Hessian's spectral norm)**.** *Consider the discriminator defined in Eq. (1). Let $H_{w_i}(D_\theta)(x)$ denote the Hessian of $D_\theta$ at $x$ with respect with the vector form of $w_i$. If the internal activations are ReLU or leaky ReLU, the spectral norm of $H_{w_i}(D_\theta)(x)$ is upper bounded by*

$$\|H_{w_i}(D_\theta)(x)\|_{sp} \leq \left\|H_{o_l^L(x)} D_\theta(x)\right\|_{sp} \cdot \|x\|^2 \cdot \prod_{i=1}^{L} \|w_i\|_{sp}^2 \bigg/ \|w_t\|_{sp}^2$$

The proof is in App. D.1. Following Prop. 3, we can easily show the upper bound of Hessian's spectral norm for spectral normalized discriminators.

**Corollary 1** (Upper bound of Hessian's spectral norm for spectral normalization)**.** *If the internal activations are ReLU or leaky ReLU, and $\|w_i\|_{sp} \leq 1$ for all $i \in [1, L]$, then*

$$\|H_{w_i}(D_\theta)(x)\|_{sp} \leq \left\|H_{o_l^L(x)} D_\theta(x)\right\|_{sp} \cdot \|x\|^2 .$$

*Moreover, if the activation for the last layer is sigmoid (e.g., for vanilla GAN [14]), we have*

$$\|H_{w_i}(D_\theta)(x)\|_{sp} \leq 0.1 \|x\|^2 ;$$

*if the activation function for the last layer is identity (e.g., for WGAN-GP [16]), we have*

$$\|H_\theta(D_\theta)(x)\|_{sp} = 0 .$$

## D.1 Proof of Prop. 3

**Lemma 1.** *The spectral norm of each internal Hessian is bounded by*

$$\left\| H_{o_a^t(x)} D_\theta(x) \right\|_{sp} \leq \left\| H_{o_l^L(x)} D_\theta(x) \right\|_{sp} \cdot \prod_{i=t+1}^L \|w_i\|_{sp}^2$$

*and*

$$\left\| H_{o_l^t(x)} D_\theta(x) \right\|_{sp} \leq \left\| H_{o_l^L(x)} D_\theta(x) \right\|_{sp} \cdot \prod_{i=t+1}^L \|w_i\|_{sp}^2$$

*Proof.* We have

$$\left\| H_{o_a^t(x)} D_\theta(x) \right\|_{sp} = \left\| w_{t+1}^T \cdot \nabla_{a_l^{t+1}(x)} D_\theta(x) \cdot w_{t+1} \right\|_{sp}$$
$$\leq \left\| \nabla_{a_l^{t+1}(x)} D_\theta(x) \right\|_{sp} \|w_{t+1}\|_{sp}^2 .$$

We also have

$$\left\| H_{o_l^t(x)} D_\theta(x) \right\|_{sp} = \left\| \text{diag}\left([a_t'(x)]_{x=o_a^t(x)}\right) \cdot H_{o_a^{t+1}(x)} D_\theta(x) \cdot \text{diag}\left([a_t'(x)]_{x=o_a^t(x)}\right) \right\|_{sp}$$
$$\leq \left\| H_{o_a^{t+1}(x)} D_\theta(x) \right\|_{sp}$$

where we use the property that ReLU or leaky ReLU is piece-wise linear. The desired inequalities then follow by induction. $\square$

Now let's come back to the proof for Prop. 3.

*Proof.* We have

$$\frac{\partial D_\theta}{\partial (w_t)_{ij} \partial (w_t)_{kl}} = \left( H_{o_l^t}(D_\theta)(x) \right)_{ik} \cdot \left( o_a^{t-1}(x) \right)_j \cdot \left( o_a^{t-1}(x) \right)_l .$$

Therefore,

$$\|H_{w_i}(D_\theta)(x)\|_{sp} \leq \left\| H_{o_l^t}(D_\theta)(x) \right\|_{sp} \left\| o_a^{t-1}(x) \right\|_\infty^2 \leq \left\| H_{o_l^t}(D_\theta)(x) \right\|_{sp} \left\| o_a^{t-1}(x) \right\|^2$$

Applying Eq. (5) and Lemma 1 we get

$$\|H_{w_i}(D_\theta)(x)\|_{sp} \leq \left\| H_{o_l^L(x)} D_\theta(x) \right\|_{sp} \cdot \prod_{i=t+1}^L \|w_i\|_{sp}^2 \cdot \|x\|^2 \cdot \prod_{i=1}^{t-1} \|w_i\|_{sp}^2$$
$$= \left\| H_{o_l^L(x)} D_\theta(x) \right\|_{sp} \cdot \|x\|^2 \cdot \prod_{i=1}^L \|w_i\|_{sp}^2 \Big/ \|w_t\|_{sp}^2$$

$\square$

# E  Proof of Thm. 1

*Proof.* For any discriminator $D_\theta = a_L \circ l_{w_L} \circ a_{L-1} \circ l_{w_{L-1}} \circ \ldots \circ a_1 \circ l_{w_1}$, consider $\theta' = \left\{ w_t' \triangleq c_t w_t \right\}_{t=1}^L$ with the constraint $\prod_{i=1}^L c_i = 1$ and $c_i \in \mathbb{R}^+$. Let $Q = \left\| \nabla_{w_i'} D_{\theta'}(x) \right\|_F \|w_i'\|_{sp}$.

We have

$$\left\| \nabla_{\theta'} D_{\theta'}(x) \right\|_{\mathrm{F}} = \sqrt{\sum_{i=1}^{L} \left\| \nabla_{w_i'} D_{\theta'}(x) \right\|_{\mathrm{F}}^2}$$

$$= \sqrt{\sum_{i=1}^{L} \frac{Q^2}{c_i^2 \left\| w_i \right\|_{\mathrm{sp}}^2}}$$

$$\geq \sqrt{L \left( \prod_{i=1}^{L} \frac{Q^2}{c_i^2 \left\| w_i \right\|_{\mathrm{sp}}^2} \right)^{1/L}}$$

$$= \sqrt{L} \cdot Q^{1/L} \cdot \left( \prod_{i=1}^{L} \left\| w_i \right\|_{\mathrm{sp}} \right)^{-1/L}$$

and the equality is achieved iff $c_i^2 \left\| w_i \right\|_{\mathrm{sp}}^2 = c_j^2 \left\| w_j \right\|_{\mathrm{sp}}^2$, $\forall i, j \in [1, L]$ according to AM-GM inequality. When $c_i^2 \left\| w_i \right\|_{\mathrm{sp}}^2 = c_j^2 \left\| w_j \right\|_{\mathrm{sp}}^2$, $\forall i, j \in [1, L]$, we have $c_t = \prod_{i=1}^{L} \left\| w_i \right\|_{\mathrm{sp}}^{1/L} \Big/ \left\| w_t \right\|_{\mathrm{sp}}$. $\qquad \square$

# F  Proof of Thm. 2

*Proof.* Since $a_{ij}$ are symmetric random variables, we know $\mathbb{E} \left( \frac{a_{ij}}{\|A\|_{\mathrm{sp}}} \right) = 0$. Further, by symmetry, we have that for any $(i, j) \neq (h, \ell)$, $\mathbb{E} \left( \frac{a_{ij}^2}{\|A\|_{\mathrm{sp}}^2} \right) = \mathbb{E} \left( \frac{a_{h\ell}^2}{\|A\|_{\mathrm{sp}}^2} \right)$. Therefore, we have

$$\mathrm{Var} \left( \frac{a_{ij}}{\|A\|_{\mathrm{sp}}} \right) = \mathbb{E} \left( \frac{a_{ij}^2}{\|A\|_{\mathrm{sp}}^2} \right) = \frac{1}{mn} \cdot \mathbb{E} \left( \frac{\sum_{i=1}^{m} \sum_{j=1}^{n} a_{ij}^2}{\|A\|_{\mathrm{sp}}^2} \right) = \frac{1}{mn} \cdot \mathbb{E} \left( \frac{\|A\|_{\mathrm{F}}^2}{\|A\|_{\mathrm{sp}}^2} \right)$$

Our approach will be to upper and lower bound the quantity $\frac{1}{mn} \cdot \mathbb{E} \left( \frac{\|A\|_{\mathrm{F}}^2}{\|A\|_{\mathrm{sp}}^2} \right)$.

**Upper bound**  Assume the singular values of $A$ are $\sigma_1 \geq \sigma_2 \geq \ldots \geq \sigma_{\min\{m,n\}}$. We have

$$\frac{1}{mn} \cdot \mathbb{E} \left( \frac{\|A\|_{\mathrm{F}}^2}{\|A\|_{\mathrm{sp}}^2} \right) = \frac{1}{mn} \cdot \mathbb{E} \left( \frac{\sum_{i=1}^{\min\{m,n\}} \sigma_i^2}{\sigma_1^2} \right) \leq \frac{\min\{m, n\}}{mn} = \frac{1}{\max\{m, n\}} \ ,$$

which gives the desired upper bound.

**Lower bound**  Now for the lower bound, if $a_{ij}$ are drawn from zero-mean Gaussian distribution and $\max\{m, n\} \geq 3$, we have

$$\frac{1}{mn} \cdot \mathbb{E} \left( \frac{\|A\|_{\mathrm{F}}^2}{\|A\|_{\mathrm{sp}}^2} \right) \tag{9}$$

$$= \frac{1}{mn} \cdot \mathbb{E} \left( \frac{1}{\|A\|_{\mathrm{sp}}^2 / \|A\|_{\mathrm{F}}^2} \right)$$

$$\geq \frac{1}{mn} \cdot \frac{1}{\mathbb{E} \left( \left\| \frac{A}{\|A\|_{\mathrm{F}}} \right\|_{\mathrm{sp}}^2 \right)}$$

$$= \frac{1}{mn} \cdot \frac{1}{\mathbb{E} \left( \|B\|_{\mathrm{sp}}^2 \right)} \tag{10}$$

where $B \in R^{m \times n}$ is uniformly sampled from the sphere of $m \times n$-dimension unit ball. We use the following lemma to lower bound (10).

**Lemma 2** (Theorem 1.1 in [44])**.** *Assume $A \in R^{m \times n}$ is uniformly sampled from the sphere of $m \times n$-dimension unit ball. When $\max\{m, n\} \geq 3$, we have*

$$\mathbb{E}\left(\|A\|_{sp}^2\right) \leq K^2\left(\mathbb{E}\left(\max_{1 \leq i \leq m} \|a_{i\bullet}\|^2\right) + \mathbb{E}\left(\max_{1 \leq j \leq n} \|a_{\bullet j}\|^2\right)\right) \;,$$

*where $K$ is a constant which does not depend on $m, n$. Here $a_{i\bullet}$ denotes the $i$-th row of $A$, and $a_{\bullet j}$ denotes the $j$-th column of $A$.*[1]

We thus have that

$$\frac{1}{mn} \cdot \frac{1}{\mathbb{E}\left(\|B\|_{\mathrm{sp}}^2\right)} \geq \frac{1}{mn} \cdot \frac{1}{K^2\left(\mathbb{E}\left(\max_{1 \leq i \leq m} \|b_{i\bullet}\|^2\right) + \mathbb{E}\left(\max_{1 \leq j \leq n} \|b_{\bullet j}\|^2\right)\right)}.$$

Hence, we need to upper bound $\mathbb{E}\left(\max_{1 \leq i \leq m} \|b_{i\bullet}\|^2\right)$ and $\mathbb{E}\left(\max_{1 \leq j \leq n} \|b_{\bullet j}\|^2\right)$. Let $z \in \mathbb{R}^m$ be a vector uniformly sampled from the sphere of $m$-dimension unit ball. Observe that $z \overset{d}{=} [\|b_{1\bullet}\|, ..., \|b_{m\bullet}\|]$. The following lemma upper bounds the square of the infinity norm of this vector.

**Lemma 3.** *Assume $z = [z_1, z_2, ..., z_n]$ is uniformly sampled from the sphere of $n$-dimension unit ball, where $n \geq 2$. Then we have*

$$\mathbb{E}\left(\max_{1 \leq i \leq n} z_i^2\right) \leq \frac{4\log(n)}{n-1}.$$

(Proof in App. F.1)

Hence, when $m, n \geq 2$, we have

$$\mathbb{E}\left(\max_{1 \leq i \leq m} \|b_{i\bullet}\|^2\right) \leq \frac{4\log(m)}{m-1}$$

Similarly, we have

$$\mathbb{E}\left(\max_{1 \leq j \leq n} \|b_{\bullet j}\|^2\right) \leq \frac{4\log(n)}{n-1}$$

Therefore,

$$\mathrm{Var}\left(\frac{a_{ij}}{\|A\|_{\mathrm{sp}}}\right)$$
$$\geq \frac{1}{mn} \cdot \frac{1}{K^2\left(\frac{4\log(m)}{m-1} + \frac{4\log(n)}{n-1}\right)}$$
$$\geq \frac{1}{8K^2} \cdot \frac{1}{n\log(m) + m\log(n)}$$
$$\geq \frac{1}{16K^2} \cdot \frac{1}{\max\{m, n\}\log(\min\{m, n\})}$$

which gives the result.

$\square$

---

[1]Note that the original theorem in [44] requires that the entries of $A$ be i.i.d. symmetric random variables, whereas in our case the entries are not i.i.d., as we require $\|A\|_{\mathrm{F}} = 1$. However, the i.i.d. assumption in their proof is only used to ensure that $A$, $S_{\sigma^{(1)}, \epsilon^{(1)}}(A)$, and $S_{\sigma^{(2)}, \epsilon^{(2)}}(A)$ have the same distribution, where $\sigma^{(t)}$ for $t = 0, 1$ are vectors of independent random permutations; $\epsilon^{(t)}$ for $t = 0, 1$ are matrices of i.i.d. random variables with equal probability of being $\pm 1$; and $S_{\sigma^{(1)}, \epsilon^{(1)}}(A) = \left(\epsilon_{ij}^{(1)} \cdot a_{i, \sigma_i^{(1)}(j)}\right)_{i,j}$ and $S_{\sigma^{(2)}, \epsilon^{(2)}}(A) = \left(\epsilon_{ij}^{(2)} \cdot a_{\sigma_j^{(2)}(i), j}\right)_{i,j}$. Our matrix $A$ satisfies this requirement, and therefore the same theorem holds.

### F.1 Proof of Lemma 3

*Proof.*

$$\mathbb{E}\left(\max_{1\leq i\leq n} z_i^2\right)$$

$$= \int_0^1 \mathbb{P}\left(\max_{1\leq i\leq n} z_i^2 \geq \delta\right) d\delta$$

$$\leq \int_0^1 \min\left\{1, n \cdot \mathbb{P}\left(z_1^2 \geq \delta\right)\right\} d\delta \tag{11}$$

where (11) follows from the union bound. Next, we use the following lemma to upper bound $\mathbb{P}\left(z_1^2 \geq \delta\right)$.

**Lemma 4.** *Assume $z = [z_1, z_2, ..., z_n]$ is uniformly sampled from the sphere of $n$-dimension unit ball, where $n \geq 2$. Then for $\frac{1}{n} \leq \delta < 1$ and $\forall i \in [1, n]$, we have*

$$\mathbb{P}\left(z_i^2 \geq \delta\right) \leq e^{-\frac{n-1}{2}\cdot\delta+1}.$$

(Proof in App. F.2). This in turn gives

$$\int_0^1 \min\left\{1, n \cdot \mathbb{P}\left(z_1^2 \geq \delta\right)\right\} d\delta \leq \int_0^{\min\left\{1, \frac{2\log(n)+2}{n-1}\right\}} 1 \cdot d\delta + \int_{\min\left\{1, \frac{2\log(n)+2}{n-1}\right\}}^1 n \cdot e^{-\frac{n-1}{2}\cdot\delta+1} \cdot d\delta \tag{12}$$

$$\leq \begin{cases} 1 & (n \leq 6) \\ \frac{2\log(n)+2}{n-1} - \frac{2n}{n-1}e^{-\frac{n-3}{2}} + \frac{2}{n-1} & (n \geq 7) \end{cases}$$

$$\leq \frac{4\log(n)}{n-1}$$

where Eq. (12) follows from Lemma 4.

$\square$

### F.2 Proof of Lemma 4

*Proof.* Due to the symmetry of $z_i$, we only need to prove the inequality for $i = 1$ case. Let $x = [x_1, ..., x_n] \sim \mathcal{N}(\mathbf{0}, I_n)$, where $I_n$ is the identity matrix in $n$ dimension. We know that $\frac{x_1^2}{\sum_{i=1}^n x_i^2} \overset{d}{=} z_1^2$. Therefore, we have

$$\mathbb{P}\left(z_1^2 \geq \delta\right) = \mathbb{P}\left(\frac{x_1^2}{\sum_{j=1}^n x_j^2} \geq \delta\right) = \mathbb{P}\left(\frac{x_1^2}{\left(\sum_{i=2}^n x_i^2\right)/(n-1)} \geq \frac{(n-1)\delta}{1-\delta}\right).$$

Note that $x_1^2$ and $\sum_{i=2}^n x_i^2$ are two independent chi-squared random variables, therefore, we know that $\frac{x_1^2}{\left(\sum_{i=2}^n x_i^2\right)/(n-1)} \sim F(1, n-1)$, where $F$ denotes the central F-distribution. Therefore,

$$\mathbb{P}\left(\frac{x_1^2}{\left(\sum_{i=2}^n x_i^2\right)/(n-1)} \geq \frac{(n-1)\delta}{1-\delta}\right) = 1 - I_\delta\left(\frac{1}{2}, \frac{n-1}{2}\right)$$

$$= I_{1-\delta}\left(\frac{n-1}{2}, \frac{1}{2}\right)$$

$$= \frac{B_{1-\delta}\left(\frac{n-1}{2}, \frac{1}{2}\right)}{B\left(\frac{n-1}{2}, \frac{1}{2}\right)}, \tag{13}$$

where $I_x(a, b)$ is the regularized incomplete beta function, $B_x(a, b)$ is the incomplete beta function, and $B(a, b)$ is beta function.

For the ease of computation, we take the $\log$ of Eq. (13). The numerator gives

$$\log\left(B_{1-\delta}\left(\frac{n-1}{2},\frac{1}{2}\right)\right)$$
$$= \log\left(\frac{(1-\delta)^{(n-1)/2}}{(n-1)/2}\,{}_2F_1\left(\frac{n-1}{2},\frac{1}{2};\frac{n+1}{2};1-\delta\right)\right)$$
$$= \frac{n-1}{2}\log\left(1-\delta\right) - \log(n-1) + \log\left({}_2F_1\left(\frac{n-1}{2},\frac{1}{2};\frac{n+1}{2};1-\delta\right)\right) + \log(2)\ , \quad (14)$$

where ${}_2F_1\left(\cdot\right)$ is the hypergeometric function. Let $(q)_i = \begin{cases} 1 & (i=0) \\ q(q+1)\dots(q+i-1) & (i>0) \end{cases}$ , we have

$$\begin{aligned}
&{}_2F_1\left(\frac{n-1}{2},\frac{1}{2};\frac{n+1}{2};1-\delta\right) \\
&= \sum_{i=0}^{\infty}\frac{\left(\frac{n-1}{2}\right)_i\left(\frac{1}{2}\right)_i(1-\delta)^i}{\left(\frac{n+1}{2}\right)_i\cdot i!} \\
&\le \sum_{i=0}^{\infty}\frac{\left(\frac{1}{2}\right)_i(1-\delta)^i}{\cdot i!} \\
&= \delta^{-\frac{1}{2}}
\end{aligned} \quad (15)$$

Substituting it into Eq. (14) gives

$$\log\left(B_{1-\delta}\left(\frac{n-1}{2},\frac{1}{2}\right)\right) \le \frac{n-1}{2}\log\left(1-\delta\right) - \log\left(n-1\right) - \frac{1}{2}\log\left(\delta\right) + \log(2)\ . \quad (16)$$

The $\log$ of the denominator of (13) is

$$\begin{aligned}
&\log\left(B\left(\frac{n-1}{2},\frac{1}{2}\right)\right) \\
&= \log\left(\frac{\Gamma\left(\frac{n-1}{2}\right)\Gamma\left(\frac{1}{2}\right)}{\Gamma\left(\frac{n}{2}\right)}\right) \\
&\ge \log\left(\sqrt{\pi}\cdot\left(\frac{n+1}{2}\right)^{-\frac{1}{2}}\right) \\
&= -\frac{1}{2}\log(n+1) + \frac{1}{2}\log(2) + \frac{1}{2}\log(\pi)\ . \quad (17)
\end{aligned}$$

where $\Gamma$ denotes the Gamma function and we use the Gautschi's inequality: $\frac{\Gamma(x+1)}{\Gamma(x+\frac{1}{2})} < (x+1)^{\frac{1}{2}}$ for positive real number $x$.

Combining Eq. (13), Eq. (16), and Eq. (17) we get

$$\begin{aligned}
&\log\left(\mathbb{P}\left(\frac{x_1^2}{\left(\sum_{i=2}^{n}x_i^2\right)/(n-1)}\ge\frac{(n-1)\,\delta}{1-\delta}\right)\right) \\
&\le \frac{n-1}{2}\log\left(1-\delta\right) - \log\left(n-1\right) + \frac{1}{2}\log(n+1) - \frac{1}{2}\log\left(\delta\right) + \frac{1}{2}\log(2/\pi) \\
&\le \frac{n-1}{2}\log\left(1-\delta\right) - \frac{1}{2}\log(n-1) - \frac{1}{2}\log(\delta) + \frac{1}{2}\log(6/\pi) \\
&\le \frac{n-1}{2}\log\left(1-\delta\right) - \frac{1}{2}\log\left(\frac{n-1}{n}\right) + \frac{1}{2}\log(6/\pi) \\
&\le \frac{n-1}{2}\log\left(1-\delta\right) + \frac{1}{2}\log\frac{12}{\pi} \\
&\le -\frac{n-1}{2}\cdot\delta + 1
\end{aligned}$$

Therefore, we have

$$\mathbb{P}\left(z_1^2 \geq \delta\right) \leq e^{-\frac{n-1}{2}\cdot\delta+1}$$

$\square$

## G   Proof of Thm. 3

*Proof.* Let $s_w = c_{in}c_{out}k_wk_h$. Since $w_{ij}$ are symmetric random variables, we know $\mathbb{E}\left(\frac{w_{ij}}{\sigma_w}\right) = 0$.
Therefore, we have

$$\text{Var}\left(\frac{w_{ij}}{\sigma_w}\right) = \mathbb{E}\left(\frac{w_{ij}^2}{\sigma_w^2}\right) = \frac{1}{s_w}\cdot\mathbb{E}\left(\frac{\sum_{i=1}^m\sum_{j=1}^n w_{ij}^2}{\sigma_w^2}\right) = \frac{1}{s_w}\cdot\mathbb{E}\left(\frac{\|w\|_F^2}{\sigma_w^2}\right)$$

Note that

$$\frac{1}{s_w}\cdot\mathbb{E}\left(\frac{\|w\|_F^2}{\sigma_w^2}\right) \in \left[\frac{2}{s_w}\cdot\mathbb{E}\left(\frac{\|w\|_F^2}{\left\|w^{c_{out}\times(c_{in}k_wk_h)}\right\|_{sp}^2 + \left\|w^{c_{in}\times(c_{out}k_wk_h)}\right\|_{sp}^2}\right),\right.$$
$$\left.\frac{4}{s_w}\cdot\mathbb{E}\left(\frac{\|w\|_F^2}{\left\|w^{c_{out}\times(c_{in}k_wk_h)}\right\|_{sp}^2 + \left\|w^{c_{in}\times(c_{out}k_wk_h)}\right\|_{sp}^2}\right)\right].$$

Assume the singular values of $w^{c_{out}\times(c_{in}k_wk_h)}$ are $\sigma_1 \geq \sigma_2 \geq \ldots \geq \sigma_{c_{out}}$, and the singular values of $w^{c_{in}\times(c_{out}k_wk_h)}$ are $\sigma'_1 \geq \sigma'_2 \geq \ldots \geq \sigma'_{c_{in}}$. We have

$$\frac{4}{s_w}\cdot\mathbb{E}\left(\frac{\|w\|_F^2}{\left\|w^{c_{out}\times(c_{in}k_wk_h)}\right\|_{sp}^2 + \left\|w^{c_{in}\times(c_{out}k_wk_h)}\right\|_{sp}^2}\right)$$
$$= \frac{4}{s_w}\cdot\mathbb{E}\left(\frac{1}{2}\cdot\frac{\sum_{i=1}^{c_{out}}\sigma_i^2}{\sigma_1^2} + \frac{1}{2}\cdot\frac{\sum_{i=1}^{c_{in}}\sigma'^2_i}{\sigma'^2_1}\right) \leq \frac{2\left(c_{out}+c_{in}\right)}{s_w} = \frac{2}{c_{in}k_wk_h + c_{out}k_wk_h},$$

which gives the desired upper bound.

As for the lower bound, observe that

$$\frac{2}{s_w}\cdot\mathbb{E}\left(\frac{\|w\|_F^2}{\left\|w^{c_{out}\times(c_{in}k_wk_h)}\right\|_{sp}^2 + \left\|w^{c_{in}\times(c_{out}k_wk_h)}\right\|_{sp}^2}\right)$$
$$= \frac{2}{s_w}\cdot\mathbb{E}\left(\frac{1}{\left\|\frac{w^{c_{out}\times(c_{in}k_wk_h)}}{\|w\|_F}\right\|_{sp}^2 + \left\|\frac{w^{c_{in}\times(c_{out}k_wk_h)}}{\|w\|_F}\right\|_{sp}^2}\right)$$
$$\geq \frac{2}{s_w}\cdot\frac{1}{\mathbb{E}\left(\left\|\frac{w^{c_{out}\times(c_{in}k_wk_h)}}{\|w\|_F}\right\|_{sp}^2\right) + \mathbb{E}\left(\left\|\frac{w^{c_{in}\times(c_{out}k_wk_h)}}{\|w\|_F}\right\|_{sp}^2\right)}$$

Then we can follow the same approach in App. F for bounding $\mathbb{E}\left(\left\|\frac{w^{c_{out}\times(c_{in}k_wk_h)}}{\|w\|_F}\right\|_{sp}^2\right)$ and $\mathbb{E}\left(\left\|\frac{w^{c_{in}\times(c_{out}k_wk_h)}}{\|w\|_F}\right\|_{sp}^2\right)$, which gives the desired lower bound. $\square$

# H Datasets and Metrics

## H.1 Datasets

**MNIST [26]**   We use the training set for our experiments, which contains 60000 images of hand-written digits of shape $28 \times 28 \times 1$. The pixels values are normalized to $[0, 1]$ before feeding to the discriminators.

**CIFAR10 [24]**   We use the training set for our experiments, which contains 50000 images of shape $32 \times 32 \times 3$. The pixels values are normalized to $[-1, 1]$ before feeding to the discriminators.

**STL10 [10]**   We use the unlabeled set for our experiments, which contains 100000 images of shape $96 \times 96 \times 3$. Following [30], we resize the images to $48 \times 48 \times 3$ for training. The pixels values are normalized to $[-1, 1]$ before feeding to the discriminators.

**CelebA [29]**   This dataset contains 202599 images. For each image, we crop the center $128 \times 128$, and resize it to $64 \times 64 \times 3$ for training. The pixels values are normalized to $[-1, 1]$ before feeding to the discriminators.

**ImageNet (ILSVRC2012) [39]**   The dataset contains 1281167 images. Following [30], for each images, we crop the central square of the images according to min(width, height), and then reshape it to $128 \times 128 \times 3$ for training. The pixels values are normalized to $[-1, 1]$ before feeding to the discriminators.

## H.2 Metrics

**Inception score [40]**   Following [30], we use 50000 generated images and split them into 10 sets for computing the score.

**FID [18]**   Following [30], we use 5000 real images and 10000 generated images for computing the score.

# I Gradient Explosion and Vanishing in GANs

## I.1 Results

To illustrate that gradient explosion and vanishing are closely related to the instability in GANs, we trained a WGAN [16] on the CIFAR10 dataset with different hyper-parameters leading to stable training, exploding gradients, and vanishing gradients over 40,000 training iterations (more experimental details in App. I.2). Fig. 10 shows the resulting inception scores for each of these runs, and Fig. 11 shows the corresponding magnitudes of the gradients over the course of training. Note that the stable run has improved sample quality and stable gradients throughout training. This phenomenon has also been observed in prior literature [2, 8]. We will demonstrate that by controlling these gradients, SN (and $SN_w$ in particular) is able to achieve more stable training and better sample quality.

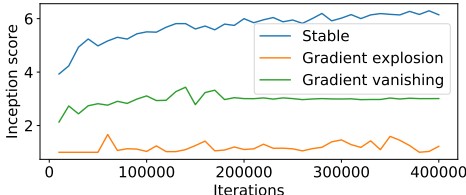

Figure 10: Inception score over the course of training. The "gradient vanishing" inception score plateaus as training is stalled.

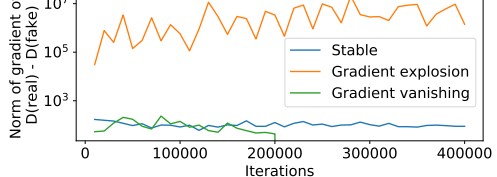

Figure 11: Norm of gradient with respect to parameters during training. The vanishing gradient collapses after 200k iterations.

| $z \in \mathbb{R}^{128} \sim \mathcal{N}(0, I)$ |
|---|
| Fully connected ($M_g \times M_g \times 512$). BN. ReLU. |
| Deconvolution ($c = 256, k = 4, s = 2$). BN. ReLU. |
| Deconvolution ($c = 128, k = 4, s = 2$). BN. ReLU. |
| Deconvolution ($c = 64, k = 4, s = 2$). BN. ReLU. |
| Deconvolution ($c = 3, k = 3, s = 1$). Tanh. |

Table 2: Generator network architectures for CIFAR10, STL10, and CelebA experiments (from [30]). For CIFAR10, $M_g = 4$. For STL10, $M_g = 6$. For CelebA, $M_g = 8$. BN stands for batch normalization. $c$ stands for number of channels. $k$ stands for kernel size. $s$ stands for stride.

| $x \in \mathbb{R}^{M \times M \times 3}$ |
|---|
| Convolution ($c = 64, k = 3, s = 1$). Leaky ReLU (0.1). |
| Convolution ($c = 64, k = 4, s = 2$). Leaky ReLU (0.1). |
| Convolution ($c = 128, k = 3, s = 1$). Leaky ReLU (0.1). |
| Convolution ($c = 128, k = 4, s = 2$). Leaky ReLU (0.1). |
| Convolution ($c = 256, k = 3, s = 1$). Leaky ReLU (0.1). |
| Convolution ($c = 256, k = 4, s = 2$). Leaky ReLU (0.1). |
| Convolution ($c = 512, k = 3, s = 1$). Leaky ReLU (0.1). |
| Fully connected (1). |

Table 3: Discriminator network architectures for CIFAR10, STL10, and CelebA experiments (from [30]). For CIFAR10, $M = 32$. For STL10, $M = 48$. For CelebA, $M = 64$. $c$ stands for number of channels. $k$ stands for kernel size. $s$ stands for stride.

## I.2 Experimental Details

The network architectures are shown in Tables 2 and 3. The dataset is CIFAR10. All experiments are run for 400k iterations. Batch size is 64. The optimizer is Adam. Let $\lambda$ be the WGAN's gradient penalty weight [16]. For the stable run, $\alpha_g = 0.0001, \alpha_d = 0.0002, \beta_1 = 0.5, \beta_2 = 0.999, \lambda = 10, n_{dis} = 1$. For the gradient explosion run, $\alpha_g = 0.001, \alpha_d = 0.001, \beta_1 = 0.5, \beta_2 = 0.999, \lambda = 10, n_{dis} = 1$. For the gradient vanishing run, $\alpha_g = 0.001, \alpha_d = 0.001, \beta_1 = 0.5, \beta_2 = 0.999, \lambda = 50, n_{dis} = 1$, and the activation functions in the discriminator are changed from leaky ReLU to ReLU.

# J   Experimental Details and Additional Results on Gradient Norms

## J.1   Experimental Details

For the MNIST experiment, the network architectures are shown in Tables 4 and 5. All experiments are run for 100 epochs. Batch size is 64. The optimizer is Adam. $\alpha_g = 0.001, \alpha_d = 0.001, \beta_1 = 0.5, \beta_2 = 0.999, n_{dis} = 1$.

For the CIFAR10 experiment, , the network architectures are shown in Tables 2 and 3. All experiments are run for 400k iterations. Batch size is 64. The optimizer is Adam. $\alpha_g = 0.0001, \alpha_d = 0.0001, \beta_1 = 0.5, \beta_2 = 0.999, n_{dis} = 1$.

Let $\lambda$ be the WGAN's gradient penalty weight [16]. For the runs without SN, $\lambda = 10$. For the runs with SN, we use the strict SN implementation [12] in order to verifying the theoretical results (the popular SN implementation [30] only gives a loose bound on the actual spectral norm of layers, see § 4). Since it already ensures that the Lipschitz constant of the discriminator is no more than 1, we discard the gradient penalty loss from training.

For all the results, the gradient norm only considers the weights and excludes the biases (if exist), so as to be consistent with the theoretical analysis.

| $z \in \mathbb{R}^{100} \sim \text{Uniform}(-1, 1)$ |
|---|
| Fully connected ($7 \times 7 \times 128$). Leaky ReLU (0.2). BN. |
| Deconvolution ($c = 64, k = 5, s = 2$). Leaky ReLU (0.2). BN. |
| Deconvolution ($c = 1, k = 5, s = 2$). Sigmoid. |

Table 4: Generator network architectures for MNIST experiments. BN stands for batch normalization. $c$ stands for number of channels. $k$ stands for kernel size. $s$ stands for stride.

| $x \in \mathbb{R}^{28 \times 28 \times 1}$ |
|---|
| Convolution ($c = 64, k = 5, s = 2$, no bias). Leaky ReLU (0.2). |
| Convolution ($c = 128, k = 5, s = 2$, no bias). Leaky ReLU (0.2). |
| Convolution ($c = 256, k = 5, s = 2$, no bias). Leaky ReLU (0.2). |
| Fully connected (1, no bias). |

Table 5: Discriminator network architectures for MNIST experiments. $c$ stands for number of channels. $k$ stands for kernel size. $s$ stands for stride.

## J.2 Additional Results

Figs. 12 and 13 show the gradient norms of each discriminator layer in MNIST and CIFAR10. Despite the difference on the network architecture and dataset, we see the similar phenomenon: when training without SN, some layers have extremely large gradient norms, which causes the overall gradient norm to be large; when training with SN, the gradient norms are much smaller and are similar across different layers.

## K  Experimental Details and Additional Results for Confirming Eq. (2)

### K.1  Experimental Details

For the MNIST experiment, the network architectures are shown in Tables 4 and 5. All experiments are run for 100 epochs. Batch size is 64. The optimizer is Adam. $\alpha_g = 0.001, \alpha_d = 0.001, \beta_1 = 0.5, \beta_2 = 0.999, n_{dis} = 1$. We use WGAN loss with the strict SN implementation [12]. Since it already ensures that the Lipschitz constant of the discriminator is no more than 1, we discard the gradient penalty loss from training. The random scaling are selected in a way the geometric mean of spectral norms of all layers equals 1.

For the CIFAR10 and STL10 experiments , the network architectures are shown in Tables 2 and 3. All experiments are run for 400k iterations. Batch size is 64. The optimizer is Adam. $\alpha_g = 0.0001, \alpha_d = 0.0001, \beta_1 = 0.5, \beta_2 = 0.999, n_{dis} = 1$. We use hinge loss [30] with the strict SN implementation [12]. The random scaling are selected in a way the geometric mean of spectral norms of all layers equals 1.75, which avoids the gradient vanishing problem as seen in § 4.

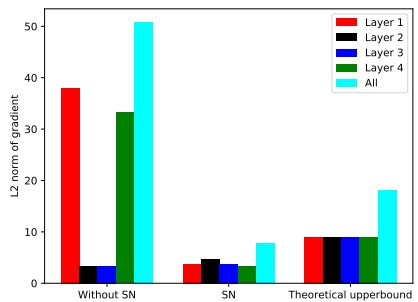

Figure 12: Gradient norms of each discriminator layer in MNIST at epoch 50.

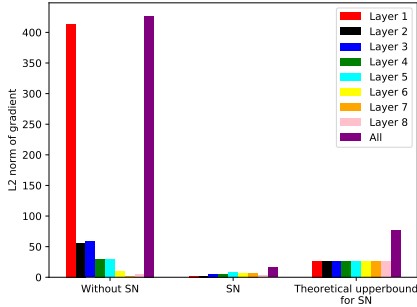

Figure 13: Gradient norms of each discriminator layer in CIFAR10 at iteration 10000.

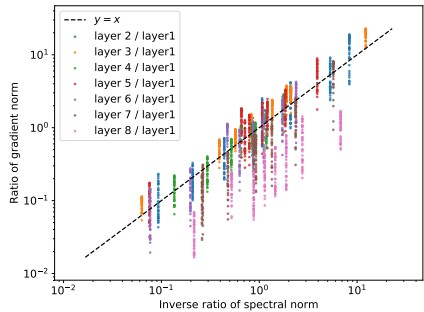
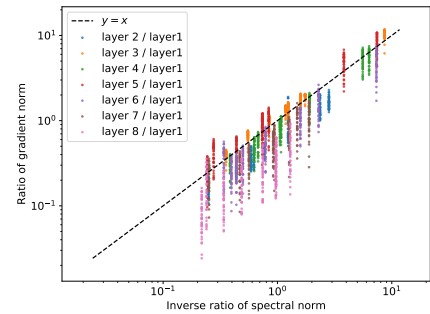

Figure 14: Ratio of gradient norm v.s. inverse ratio of spectral norm in CIFAR10.

Figure 15: Ratio of gradient norm v.s. inverse ratio of spectral norm in STL10.

### K.2 Additional Results

Figs. 14 and 15 show the ratios of the gradient norms at each layer and the inverse ratios of the spectral norms in CIFAR10 and STL10. Generally, we see that the most of the points are near the diagonal line, which means that the assumption in Eq. (2) is reasonably true in practice. However, we note that the last layer (layer 8) somehow has slightly smaller gradient, as the points of "layer 8 / layer 1" are slightly lower than the diagonal line. This could result from the fact that layer 8 is a fully connected layer whereas all other layers are convolutional layers. We defer the more detailed analysis of this phenomenon to future work.

## L  Experimental Details and Additional Results on Vanishing Gradient

### L.1  Experimental Details

The network architectures are shown in Tables 2 and 3. The dataset is CIFAR10. All experiments are run for 400k iterations. Batch size is 64. The optimizer is Adam. $\alpha_g = 0.0001, \alpha_d = 0.0001, \beta_1 = 0.5, \beta_2 = 0.999, n_{dis} = 1$. We use hinge loss [30].

### L.2  Parameter Variance With and Without SN

Figs. 16 and 17 show the parameter variance of each layer without and with SN. Note that Fig. 17 is just collecting the empirical lines in Fig. 7 for the ease of comparison here. Figs. 18 and 19 show the gradient norm and inception score.

We can see that when training with SN, the parameter variance is stable throughout training (Fig. 17), and the magnitude of gradient is also stable (Fig. 18) . However, when training without SN, the parameter variance tends to increase throughout training (Fig. 16), which causes a quick decrease in the magnitude of gradient in the begining of training (Fig. 18) because of the saturation of hinge loss (§ 4). Because SN promotes the stability of the variance and gradient throughout training, we see that SN improves the sample quality significantly (Fig. 19).

### L.3  Comparing Two Variants Spectral Norms

Figs. 20 and 21 show the ratio between two versions of spectral norm [30, 12] throughout the training of the popular SN [30] and the strict SN [12]. $\|Conv\|_{sp}$ denotes the spectral norm of the expanded matrix $\|\tilde{w}\|_{sp}$ used in [12]. $\|w\|_{sp}$ denotes the spectral norm of reshaped matrix $\|\hat{w}\|_{sp}$ used in [30]. The theoretical lower and upper bound are calculated according to Corollary 1 in [47]. We can see that no matter in which architecture, $\|\tilde{w}\|_{sp}$ is usually strictly larger than $\|\hat{w}\|_{sp}$. Note that the reason why in some cases the ratio exceeds the upper bound in Fig. 20 is because the spectral norms are calculated using power iteration [30, 12] which has approximation error.

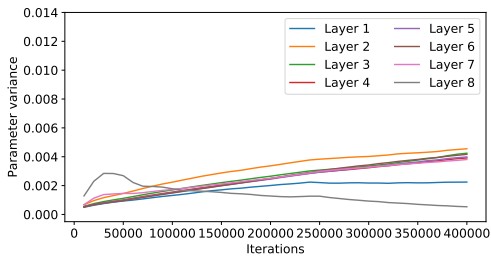
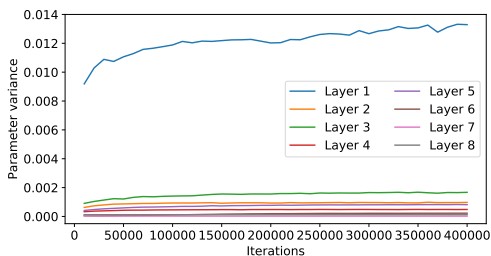

Figure 16: Parameter variance without SN in CIFAR10.

Figure 17: Parameter variance with SN in CIFAR10.

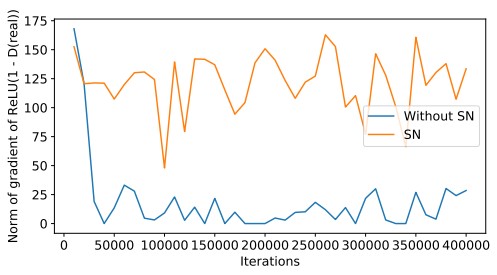
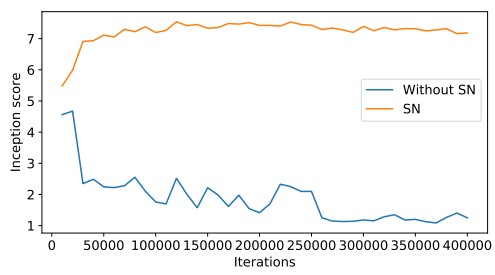

Figure 18: Gradient norm with and without SN in CIFAR10.

Figure 19: Inception score with and without SN in CIFAR10.

### L.4   Parameter Variance of Scaled SN

Figure Fig. 22 shows the parameter variance of scaled SN for both SN versions [30, 12]. We can see that when scale=1.75, the product of parameter variances for $SN_{Conv}$ [12] is similar to the one of $SN_w$ [30]. Moreover, by comparing Fig. 22 and Fig. 6 we can see that when the products of variances of two SN variants are similar, the sample quality is also similar. This confirms the intuition from LeCun initialization [25] that the magnitude of variance plays an important role on the performance of neural network, and it should not be too large nor too small.

## M   Experimental Details and Additional Results on Scaling (§ 5.2)

### M.1   Experimental Details

The network architectures are shown in Tables 2 and 3. SN models are run for 400k iterations. LeCun initialization models are run till the sample quality converges or starts dropping (usually within 400k iterations). Batch size is 64. The optimizer is Adam. $\alpha_g = 0.0001, \alpha_d = 0.0001, \beta_1 = 0.5, \beta_2 = 0.999, n_{dis} = 1$. We use hinge loss [30].

Since LeCun initialization is unstable when the scaling is not proper, in Fig. 8, we plot the best score during training instead of the score at the end of training.

### M.2   Additional Results

Although the good scaling modes for SN and LeCun initialization seem to be very different in Fig. 8, there indeed exists a (perhaps coincidental) correspondence in terms of parameter variances. In Fig. 23, we show the inception score v.s. parameter variances for SN and LeCun initialization. We can see that the first good mode occurs when log of the product of parameter variances is around -70 to -60, and the second mode is around -50 to -40.

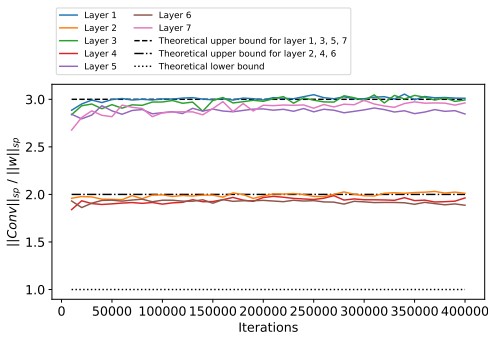
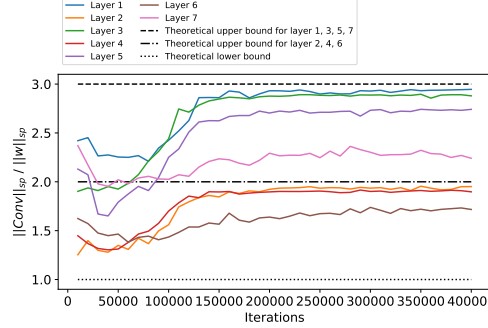

Figure 20: The ratio of two spectral norms throughout the training of the popular SN [30] in CIFAR10.

Figure 21: The ratio of two spectral norms throughout the training of the strict SN [12] in CIFAR10.

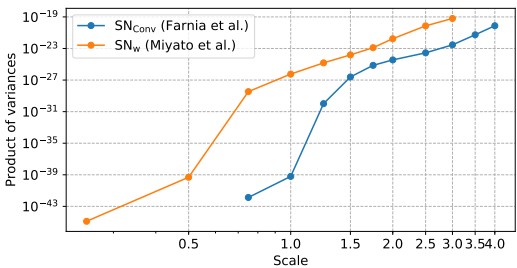

Figure 22: The parameter variance of scaled SN in CIFAR10.

## N    Results with Different Hyper-parameters and SN Variants

In addition to SN [30], we compare against two variants of SN proposed in the appendix of [30], which we denote "same $\gamma$" and "diff. $\gamma$" (details in App. O). These two variants are reported to be worse than SN in [30] and are not used in practice, but we include them here for reference. We run experiments on CIFAR10, STL10, CelebA, and ImageNet, with two widely-used metrics for sample quality: inception score [40] and Frechet Inception Distance (FID) [18] (details in App. H).

We use the network architecture from SN [30]. We controlled five hyper-parameters (Table 7, App. P): $\alpha_g$ and $\alpha_d$, the generator/discriminator learning rates, $\beta_1, \beta_2$, Adam momentum parameters [23], and $n_{dis}$, the number of discriminator updates per generator update. Three hyper-parameter settings are from [30], with equal discriminator and generator learning rates; the final two test *unequal* learning rates for showing a more thorough comparison. More details are in Apps. P and Q.

As in [30], we report the metrics from the best hyper-parameter for each algorithm in Table 6. *BSN outperforms the standard SN in all sample quality metrics except FID score on STL10, where their metrics are within standard error of each other.* Regarding the SN variants with $\gamma$, in CIFAR10 and STL10, they have worse performance than SN and BSN, same as reported in [30]. In CelebA, the SN variants have better performance for the best hyper-parameter setting. But in general, these SN variants are very sensitive to hyper-parameters (Apps. P to R), therefore they are not adopted in practice [30]. Nevertheless, BSN is still able to improve or have similar performance on those two variants in most of the settings.

More importantly, the superiority of BSN is stable across hyper-parameters. Figs. 24 and 25 show the inception scores of all the hyper-parameters we tested on CIFAR10 and STL10. BSN has the best or competitive performance in most of the settings. The only exception is $n_{dis} = 5$ setting in STL10, where we observe that the performance from both SN and BSN have larger variance across different random seeds, and the SN variants with $\gamma$ perform better. On CelebA, BSN also outperforms SN in FID across all hyper-parameters (App. R), and it outperforms all SN variants in every hyper-parameter setting except one (Fig. 55).

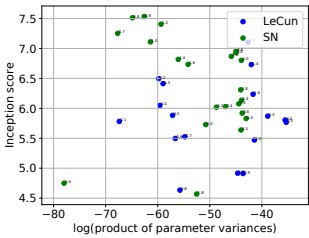

Figure 23: Inception score v.s. parameter variances of scaled SN and scaled LeCun initialization in CIFAR10. Each point corresponds to one run, at the point when the score is the best during training. The numbers near each point indicate the scaling.

| | CIFAR10 | | STL10 | | CelebA |
|---|---|---|---|---|---|
| | Inception score ↑ | FID ↓ | Inception score ↑ | FID ↓ | FID ↓ |
| Real data | 11.26 | 9.70 | 26.70 | 10.17 | 4.44 |
| SN (same $\gamma$) | $6.46 \pm 0.06$ | $42.35 \pm 0.74$ | $\mathbf{8.86 \pm 0.03}$ | $\mathbf{54.61 \pm 0.51}$ | $\mathbf{7.74 \pm 0.11}$ |
| BSN (same $\gamma$) | $\mathbf{6.69 \pm 0.05}$ | $\mathbf{39.62 \pm 0.40}$ | $8.76 \pm 0.03$ | $55.04 \pm 0.48$ | $7.83 \pm 0.09$ |
| SN (diff. $\gamma$) | $6.53 \pm 0.01$ | $41.88 \pm 0.50$ | $8.79 \pm 0.03$ | $56.76 \pm 0.44$ | $\color{red}{\mathbf{7.54 \pm 0.08}}$ |
| BSN (diff. $\gamma$) | $\mathbf{6.72 \pm 0.05}$ | $\mathbf{38.15 \pm 0.72}$ | $\mathbf{8.80 \pm 0.03}$ | $\mathbf{53.99 \pm 0.33}$ | $7.67 \pm 0.04$ |
| SN | $7.22 \pm 0.09$ | $31.43 \pm 0.90$ | $9.16 \pm 0.03$ | $\color{red}{\mathbf{42.89 \pm 0.54}}$ | $9.09 \pm 0.32$ |
| BSN | $\color{red}{\mathbf{7.58 \pm 0.04}}$ | $\color{red}{\mathbf{26.62 \pm 0.21}}$ | $\mathbf{9.25 \pm 0.01}$ | $42.98 \pm 0.54$ | $\mathbf{8.54 \pm 0.20}$ |

Table 6: Inception scores and FIDs on CIFAR10, STL10, and CelebA. Each experiment is conducted with 5 random seeds, with mean and standard error reported. We follow the common practice of excluding Inception Score in CelebA as the inception network is pretrained on ImageNet, which is very different from CelebA. The bold font marks the best numbers between SN and BSN using the same variant. The red color marks the best numbers among all runs. The "same $\gamma$" and "diff. $\gamma$" variants are not used in practice and are reported to have bad performance in [30].

More results (generated images, training curves, FID plots) are in Apps. P to R.

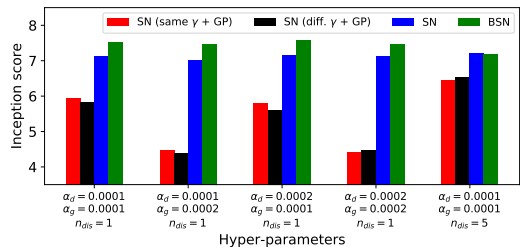

Figure 24: Inception score in CIFAR10. The results are averaged over 5 random seeds.

## O    Details on SN Variants

In Appendix E of [30], a variant of SN is introduced. Instead of strictly setting the spectral norm of each layer, the idea of this approach is to release the constraint by multiplying each spectral normalized weights with a trainable parameter $\gamma$. However, this would make the gradient of discriminator arbitrarily large, which violates the original motivation of SN. Therefore, the approach incorporates gradient penalty [16] for setting the Lipschitz constant of discriminator to 1. The gradient penalty weights are set to 10 in all experiments.

However, from the description in [30], it is unclear if all layers have the same or separated $\gamma$. Therefore, we try both versions in our experiments. "Same $\gamma$" denotes that version where all layers share the same $\gamma$. "Diff. $\gamma$" denotes the version where each layer has a separate $\gamma$.

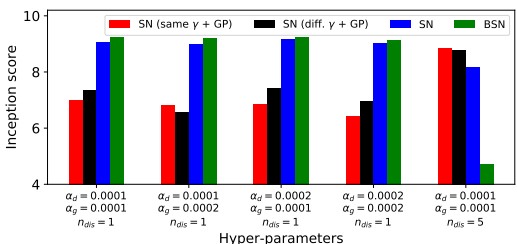

Figure 25: Inception score in STL10. The results are averaged over 5 random seeds.

| $\alpha_g$ | $\alpha_d$ | $\beta_1$ | $\beta_2$ | $n_{dis}$ |
|---|---|---|---|---|
| 0.0001 | 0.0001 | 0.5 | 0.9 | 5 |
| 0.0001 | 0.0001 | 0.5 | 0.999 | 1 |
| 0.0002 | 0.0002 | 0.5 | 0.999 | 1 |
| 0.0001 | 0.0002 | 0.5 | 0.999 | 1 |
| 0.0002 | 0.0001 | 0.5 | 0.999 | 1 |

Table 7: Hyper-parameters tested in CIFAR10 and STL10 experiments. The first three settings are from [30, 16, 49, 38]. $\alpha_g$ and $\alpha_d$: learning rates for generator and discriminator. $\beta_1, \beta_2$: momentum parameters in Adam. $n_{dis}$: number of discriminator updates per generator update.

# P  Experimental Details and Additional Results on CIFAR10

## P.1  Experimental Details

The network architectures are shown in Tables 2 and 3. All experiments are run for 400k iterations. Batch size is 64. The optimizer is Adam. We use the five hyper-parameter settings listed in Table 7. (In Table 1 we only show the results from the first hyper-parameter setting.) We use hinge loss with the popular SN implementation [30].

For SSN in Table 1, we ran following scales: [0.7, 0.8, 0.9, 1.0, 1.2, 1.4, 1.6, 1.8, 2.0, 2.2, 2.4, 2.6, 2.8, 3.0, 3.2, 3.4, 3.6, 3.8, 4.0, 4.5, 5.0, 5.5, 6.0, 7.0, 8.0, 9.0, 10.0], and present the results from best one for each metric. For BSSN in Table 1, we ran the following scales: [0.7, 0.8, 0.9, 1.0, 1.2, 1.4, 1.6, 1.8, 2.0, 2.2, 2.4, 2.6, 2.8, 3.0, 3.2, 3.4, 3.6, 3.8, 4.0], and present the results from the best one for each metric.

## P.2  FID Plot

Fig. 26 shows the FID score in CIFAR10 dataset. We can see that BSN has the best performance in all 5 hyper-parameter settings.

## P.3  Training Curves

From App. N we can see that SN (no $\gamma$) and BSN generally have the best performance. Therefore, in this section, we focus on comparing these two algorithms with the training curves. Figs. 9 and 27

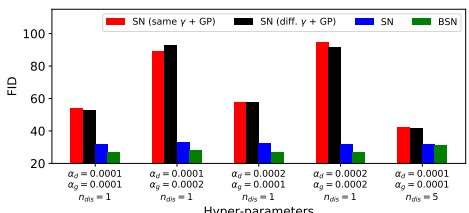

Figure 26: FID in CIFAR10. The results are averaged over 5 random seeds.

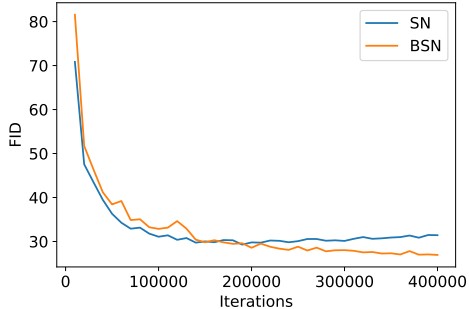

Figure 27: FID in CIFAR10. The results are averaged over 5 random seeds. The hyper-parameters are: $\alpha_g = 0.0001$, $\alpha_d = 0.0001$, $n_{dis} = 1$.

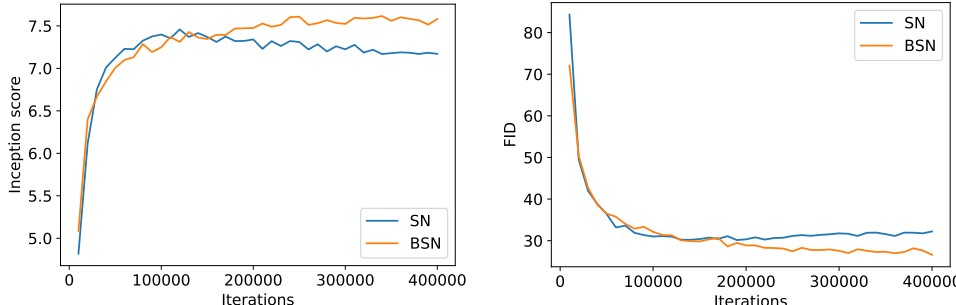

Figure 28: Inception score in CIFAR10. The results are averaged over 5 random seeds. The hyper-parameters are: $\alpha_g = 0.0001$, $\alpha_d = 0.0002$, $n_{dis} = 1$.

Figure 29: FID in CIFAR10. The results are averaged over 5 random seeds. The hyper-parameters are: $\alpha_g = 0.0001$, $\alpha_d = 0.0002$, $n_{dis} = 1$.

to 35 show the inception score and FID of these two algorithms during training. Generally, we see that BSN converges slower than SN *at the beginning of training*. However, as training proceeds, the sample quality of SN often drops (e.g. Figs. 9 and 27 to 33), whereas the sample quality of BSN always increases and then stabilizes at the high level. In most cases, BSN not only outperforms SN at the end of training, but also outperforms the peak sample quality of SN during training (e.g. Figs. 9 and 27 to 33). From these results, we can conclude that BSN improves both the sample quality and training stability over SN.

## P.4 Generated Images

Figs. 36 to 39 show the generated images from the run with the best inception score for each algorithm.

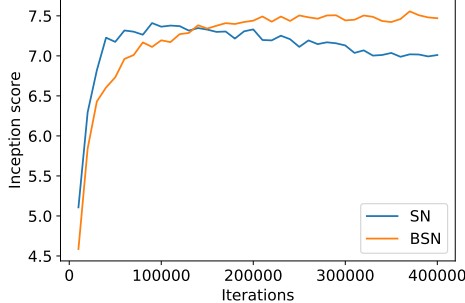

Figure 30: Inception score in CIFAR10. The results are averaged over 5 random seeds. The hyper-parameters are: $\alpha_g = 0.0002$, $\alpha_d = 0.0001$, $n_{dis} = 1$.

Figure 31: FID in CIFAR10. The results are averaged over 5 random seeds. The hyper-parameters are: $\alpha_g = 0.0002$, $\alpha_d = 0.0001$, $n_{dis} = 1$.

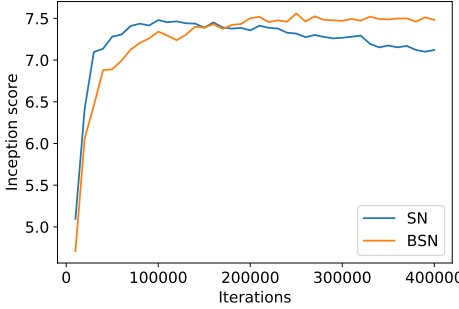
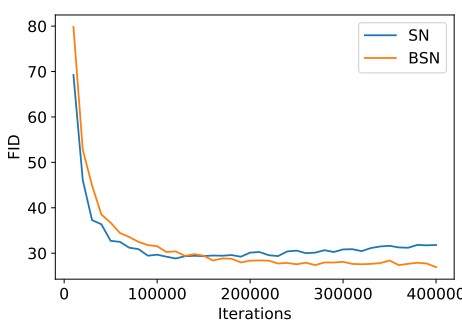

Figure 32: Inception score in CIFAR10. The results are averaged over 5 random seeds. The hyper-parameters are: $\alpha_g = 0.0002$, $\alpha_d = 0.0002$, $n_{dis} = 1$.

Figure 33: FID in CIFAR10. The results are averaged over 5 random seeds. The hyper-parameters are: $\alpha_g = 0.0002$, $\alpha_d = 0.0002$, $n_{dis} = 1$.

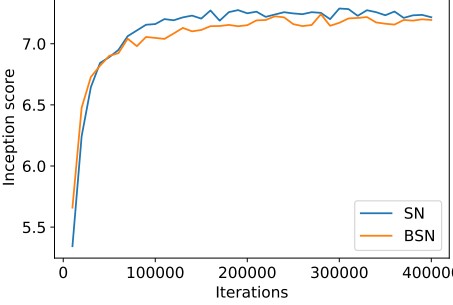
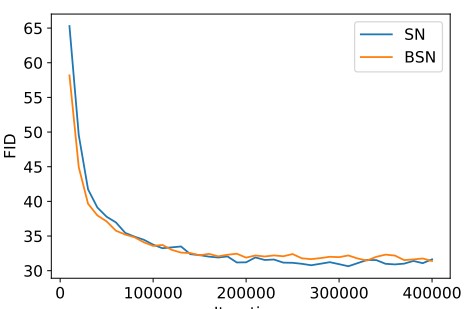

Figure 34: Inception score in CIFAR10. The results are averaged over 5 random seeds. The hyper-parameters are: $\alpha_g = 0.0001$, $\alpha_d = 0.0001$, $n_{dis} = 5$.

Figure 35: FID in CIFAR10. The results are averaged over 5 random seeds. The hyper-parameters are: $\alpha_g = 0.0001$, $\alpha_d = 0.0001$, $n_{dis} = 5$.

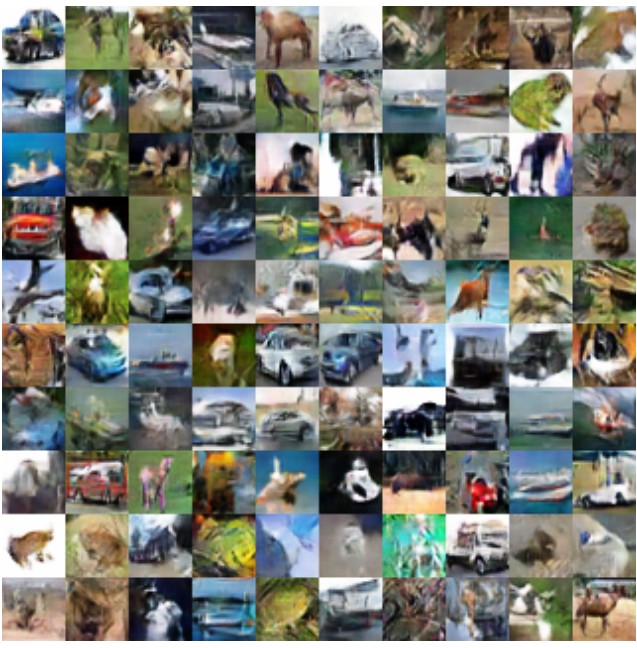

Figure 36: Generated samples from the best run of SN (same $\gamma$) in CIFAR10. The hyper-parameters are: $\alpha_g = 0.0001$, $\alpha_d = 0.0001$, $n_{dis} = 5$. Inception score is 6.64. FID is 41.01.

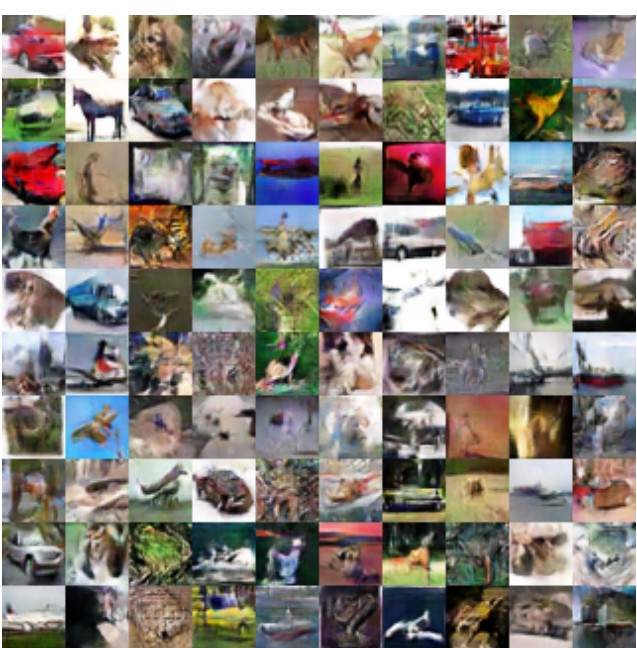

Figure 37: Generated samples from the best run of SN (diff. $\gamma$) in CIFAR10. The hyper-parameters are: $\alpha_g = 0.0001$, $\alpha_d = 0.0001$, $n_{dis} = 5$. Inception score is 6.55. FID is 41.18.

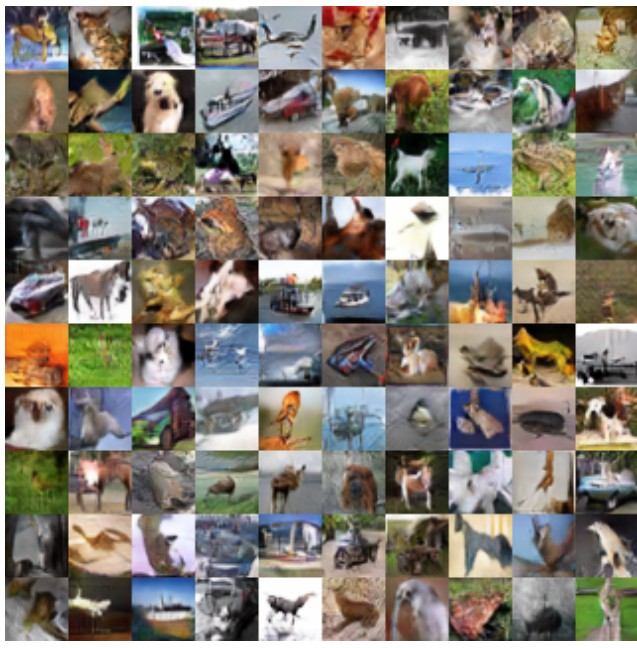

Figure 38: Generated samples from the best run of SN in CIFAR10. The hyper-parameters are: $\alpha_g = 0.0001$, $\alpha_d = 0.0002$, $n_{dis} = 1$. Inception score is 7.56. FID is 28.64.

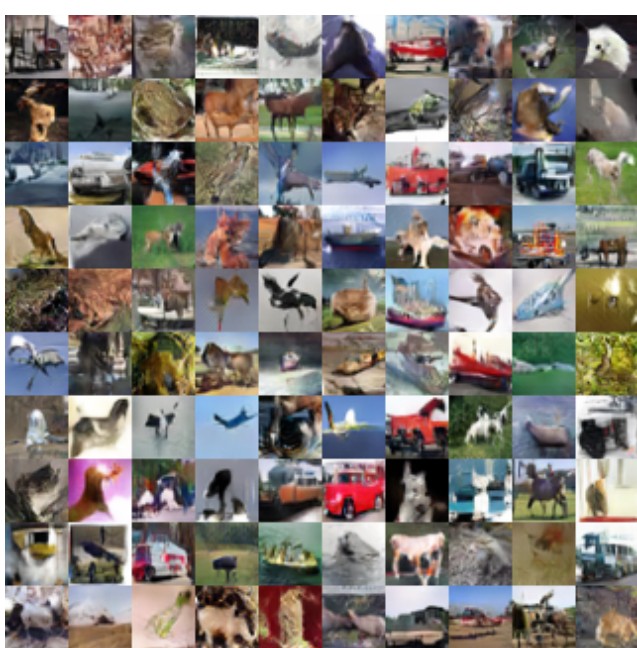

Figure 39: Generated samples from the best run of BSN in CIFAR10. The hyper-parameters are: $\alpha_g = 0.0001$, $\alpha_d = 0.0002$, $n_{dis} = 1$. Inception score is 7.70. FID is 25.96.

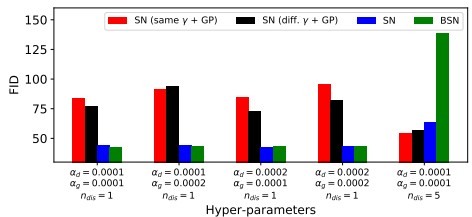

Figure 40: FID in STL10. The results are averaged over 5 random seeds.

## Q Experimental Details and Additional Results on STL10

### Q.1 Experimental Details

The network architectures are shown in Tables 2 and 3. Batch size is 64. The optimizer is Adam. We use the five hyper-parameter settings listed in Table 7. (In Table 1 we only show the results from the first hyper-parameter setting.) We use hinge loss with the popular SN implementation [30].

SN (no $\gamma$) and BSN under $n_{dis} = 1$ settings are run for 800k iterations as we observe that they need longer time to converge. All other experiments are run for 400k iterations.

For SSN and BSSN in Table 1, we ran following scales: [0.7, 0.8, 0.9, 1.0, 1.2, 1.4, 1.6], and present the results from best one for each metric.

### Q.2 FID Plot

Fig. 40 shows the FID score in STL10 dataset. We can see that BSN has the best or competitive performance in most of the hyper-parameter settings. Again, the only exception is $n_{dis} = 5$ setting.

### Q.3 Training Curves

From App. N we can see that SN (no $\gamma$) and BSN generally have the best performance. Therefore, in this section, we focus on comparing these two algorithms with the training curves. Figs. 41 to 50 show the inception score and FID of these two algorithms during training. Generally, we see that BSN converges slower than SN *at the beginning of training*. However, as training proceeds, BSN finally has better metrics in most cases. Note that unlike CIFAR10, SN seems to be more stable in STL10 as its sample quality does not drop in most hyper-parameters. But the key conclusion is the same: in most cases, BSN not only outperforms SN at the end of training, but also outperforms the peak sample quality of SN during training (e.g. Figs. 41 to 48). The only exception is the $n_{dis} = 5$ setting, where both SN and BSN has instability issue: the sample quality first improves and then significantly drops. The problem with BSN seems to be severer. We discussed about this problem in App. N.

### Q.4 Generated Images

Figs. 51 to 54 show the generated images from the run with the best inception score for each algorithm.

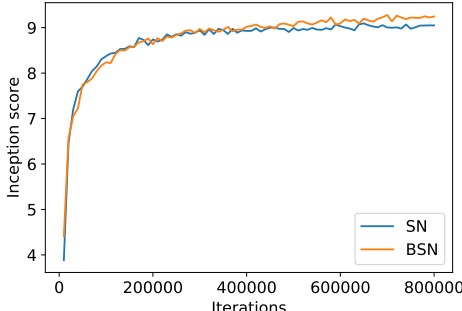

Figure 41: Inception score in STL10. The results are averaged over 5 random seeds. The hyper-parameters are: $\alpha_g = 0.0001$, $\alpha_d = 0.0001$, $n_{dis} = 1$.

Figure 42: FID in STL10. The results are averaged over 5 random seeds. The hyper-parameters are: $\alpha_g = 0.0001$, $\alpha_d = 0.0001$, $n_{dis} = 1$.

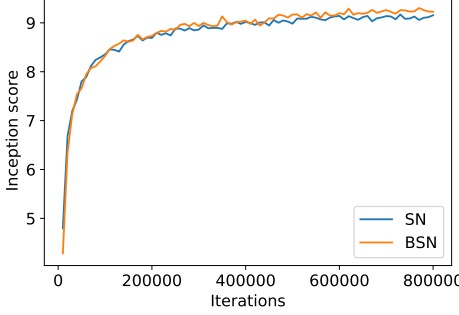
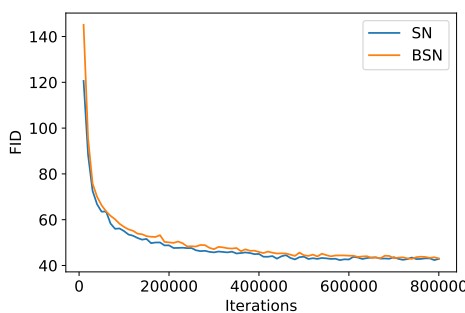

Figure 43: Inception score in STL10. The results are averaged over 5 random seeds. The hyper-parameters are: $\alpha_g = 0.0001$, $\alpha_d = 0.0002$, $n_{dis} = 1$.

Figure 44: FID in STL10. The results are averaged over 5 random seeds. The hyper-parameters are: $\alpha_g = 0.0001$, $\alpha_d = 0.0002$, $n_{dis} = 1$.

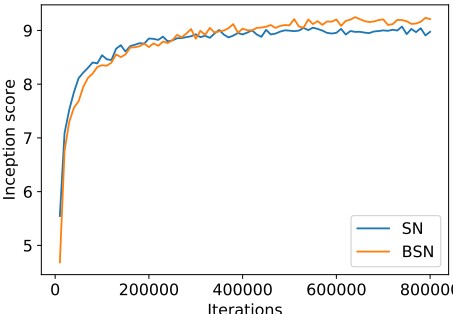
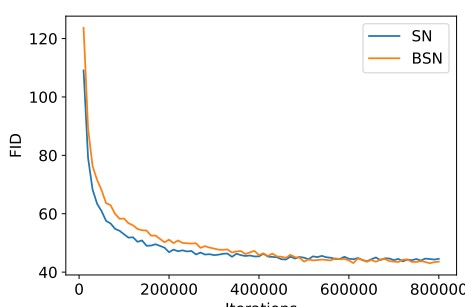

Figure 45: Inception score in STL10. The results are averaged over 5 random seeds. The hyper-parameters are: $\alpha_g = 0.0002$, $\alpha_d = 0.0001$, $n_{dis} = 1$.

Figure 46: FID in STL10. The results are averaged over 5 random seeds. The hyper-parameters are: $\alpha_g = 0.0002$, $\alpha_d = 0.0001$, $n_{dis} = 1$.

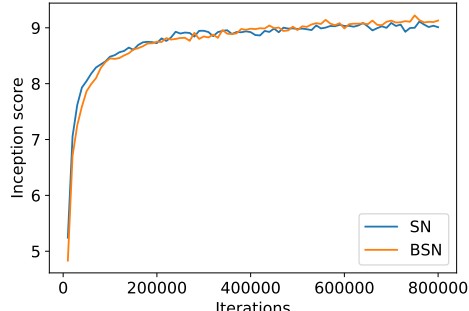

Figure 47: Inception score in STL10. The results are averaged over 5 random seeds. The hyper-parameters are: $\alpha_g = 0.0002$, $\alpha_d = 0.0002$, $n_{dis} = 1$.

Figure 48: FID in STL10. The results are averaged over 5 random seeds. The hyper-parameters are: $\alpha_g = 0.0002$, $\alpha_d = 0.0002$, $n_{dis} = 1$.

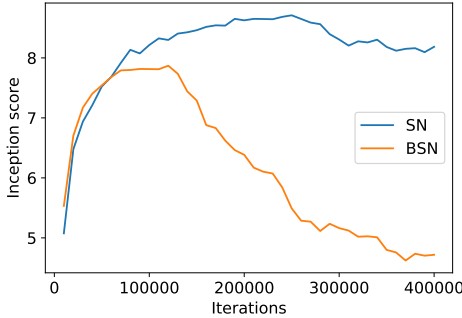

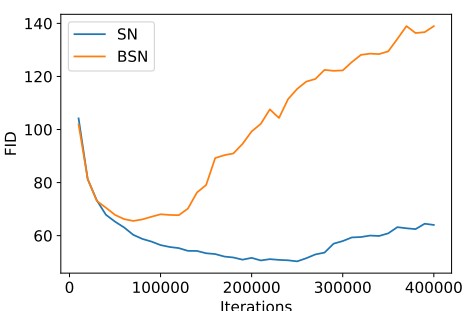

Figure 49: Inception score in STL10. The results are averaged over 5 random seeds. The hyper-parameters are: $\alpha_g = 0.0001$, $\alpha_d = 0.0001$, $n_{dis} = 5$.

Figure 50: FID in STL10. The results are averaged over 5 random seeds. The hyper-parameters are: $\alpha_g = 0.0001$, $\alpha_d = 0.0001$, $n_{dis} = 5$.

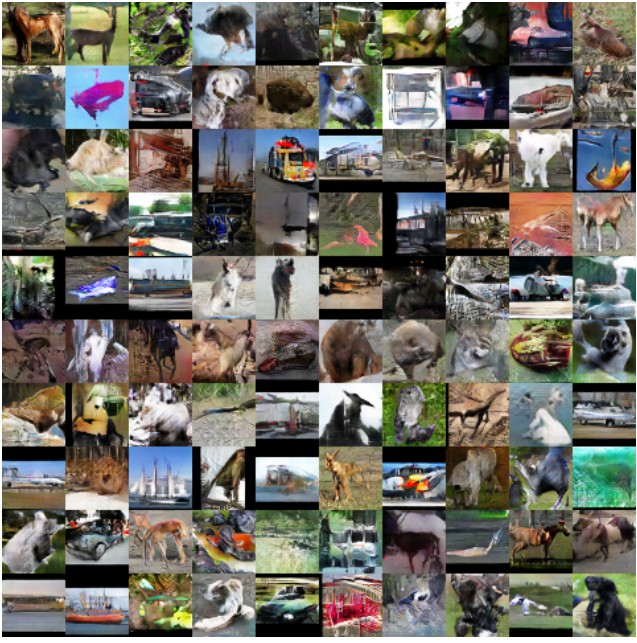

Figure 51: Generated samples from the best run of SN (same $\gamma$) in STL10. The hyper-parameters are: $\alpha_g = 0.0001$, $\alpha_d = 0.0001$, $n_{dis} = 5$. Inception score is $8.96$. FID is $53.94$.

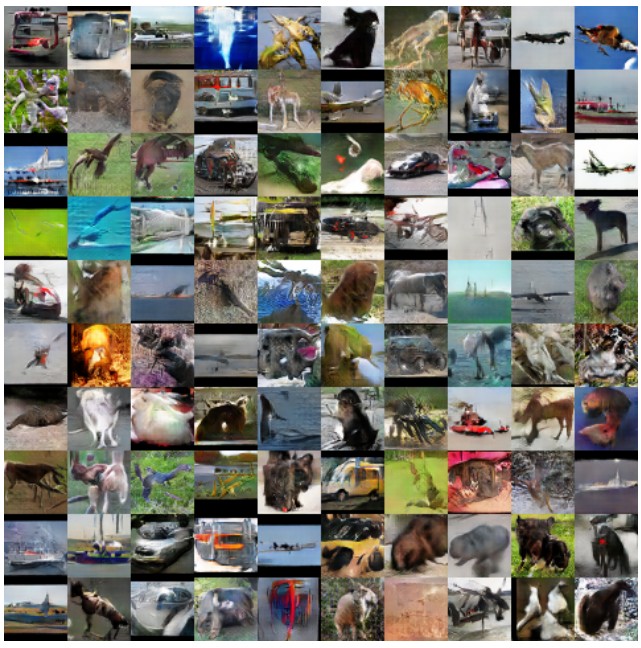

Figure 52: Generated samples from the best run of SN (diff. $\gamma$) in STL10. The hyper-parameters are: $\alpha_g = 0.0001$, $\alpha_d = 0.0001$, $n_{dis} = 5$. Inception score is 8.88. FID is 56.14.

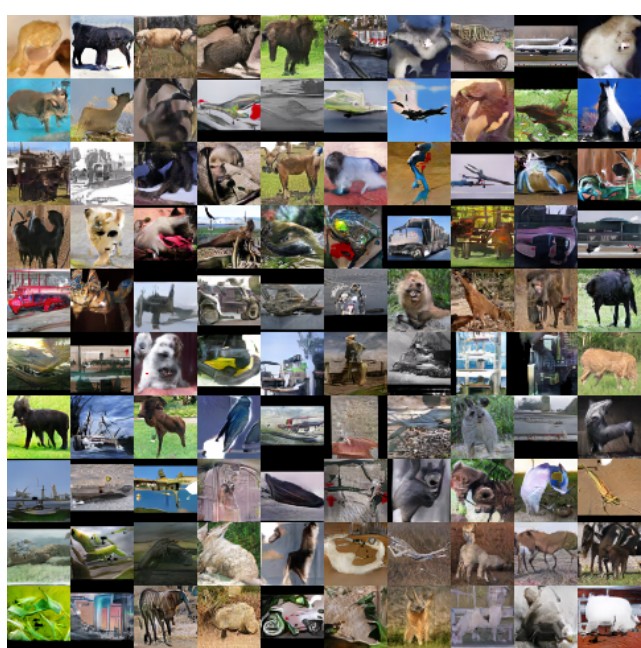

Figure 53: Generated samples from the best run of SN in STL10. The hyper-parameters are: $\alpha_g = 0.0001$, $\alpha_d = 0.0002$, $n_{dis} = 1$. Inception score is 9.26. FID is 44.38.

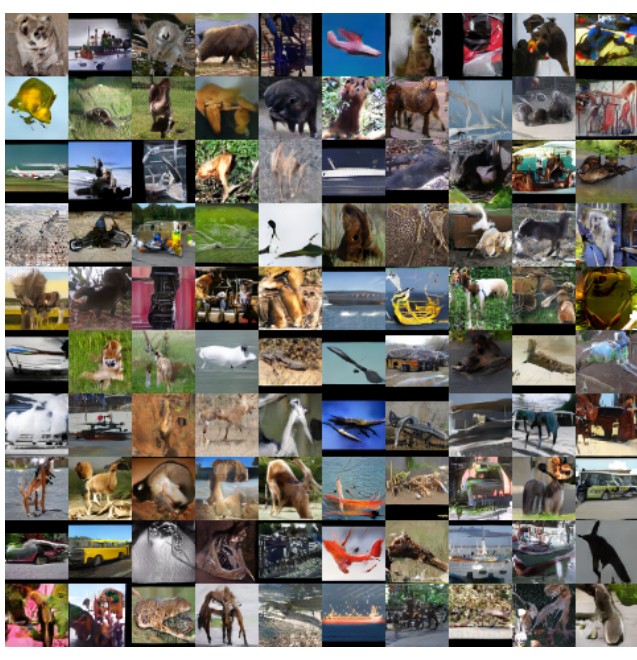

Figure 54: Generated samples from the best run of BSN in STL10. The hyper-parameters are: $\alpha_g = 0.0001$, $\alpha_d = 0.0002$, $n_{dis} = 1$. Inception score is 9.46. FID is 42.78.

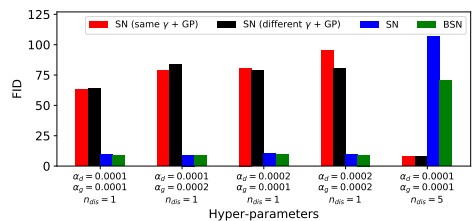

Figure 55: FID in CelebA. The results are averaged over 5 random seeds.

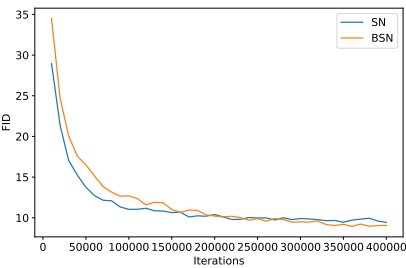

Figure 56: FID in CelebA. The results are averaged over 5 random seeds. The hyper-parameters are: $\alpha_g = 0.0001$, $\alpha_d = 0.0001$, $n_{dis} = 1$.

## R Experimental Details and Additional Results on CelebA

### R.1 Experimental Details

The network architectures are shown in Tables 2 and 3. All experiments are run for 400k iterations. Batch size is 64. The optimizer is Adam. We use the five hyper-parameter settings listed in Table 7. (In Table 1 we only show the results from the first hyper-parameter setting.) We use hinge loss with the popular SN implementation [30].

For SSN and BSSN in Table 1, we ran following scales: [0.7, 0.8, 0.9, 1.0, 1.2, 1.4, 1.6], and present the results from best one for each metric.

### R.2 FID Plot

Fig. 55 shows the FID score in CelebA dataset. We can see that BSN outperforms the standard SN in all 5 hyper-parameter settings.

### R.3 Training Curves

From App. N we can see that SN (no $\gamma$) and BSN generally have the best performance. Therefore, in this section, we focus on comparing these two algorithms with the training curves. Figs. 56 to 60 show the FID of these two algorithms during training. Generally, we see that BSN converges slower than SN *at the beginning of training*. However, as training proceeds, BSN finally has better metrics in all cases. Note that unlike CIFAR10, SN seems to be more stable in CelebA as its sample quality does not drop in most hyper-parameters. But the key conclusion is the same: in most cases, BSN not only outperforms SN at the end of training, but also outperforms the peak sample quality of SN during training (e.g. Figs. 56 to 59). The only exception is the $n_{dis} = 5$ setting, where both SN and BSN has instability issue: the sample quality first improves and then significantly drops. But even in this case, BSN has better final performance than the standard SN.

### R.4 Generated Images

Figs. 61 to 64 show the generated images from the run with the best FID for each algorithm.

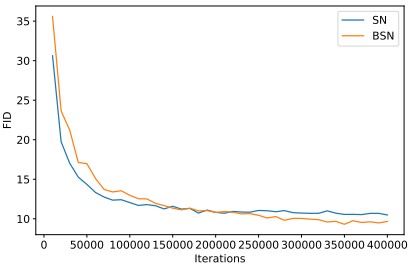

Figure 57: FID in CelebA. The results are averaged over 5 random seeds. The hyper-parameters are: $\alpha_g = 0.0001$, $\alpha_d = 0.0002$, $n_{dis} = 1$.

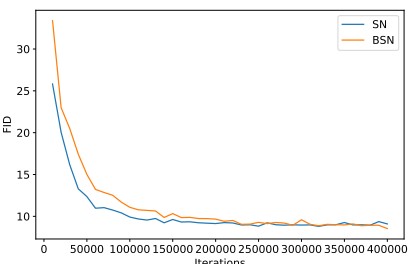

Figure 58: FID in CelebA. The results are averaged over 5 random seeds. The hyper-parameters are: $\alpha_g = 0.0002$, $\alpha_d = 0.0001$, $n_{dis} = 1$.

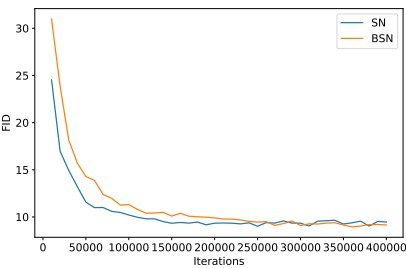

Figure 59: FID in CelebA. The results are averaged over 5 random seeds. The hyper-parameters are: $\alpha_g = 0.0002$, $\alpha_d = 0.0002$, $n_{dis} = 1$.

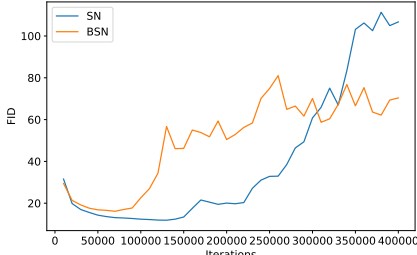

Figure 60: FID in CelebA. The results are averaged over 5 random seeds. The hyper-parameters are: $\alpha_g = 0.0001$, $\alpha_d = 0.0001$, $n_{dis} = 5$.

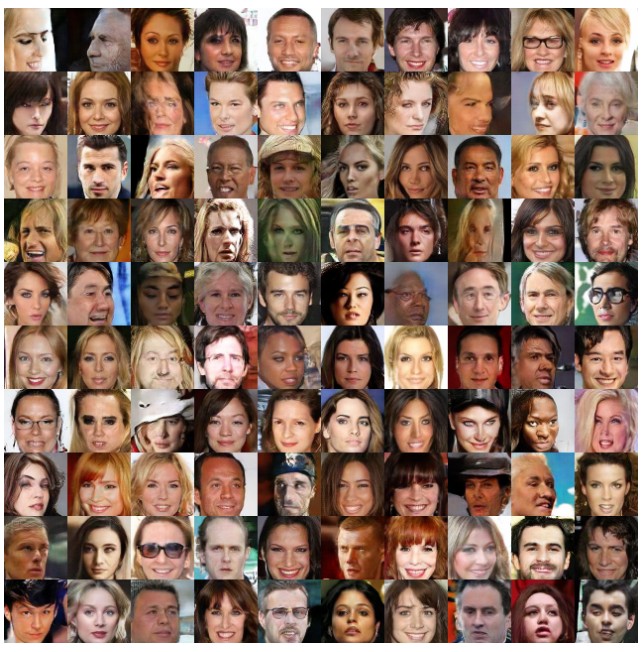

Figure 61: Generated samples from the best run of SN (same $\gamma$) in CelebA. The hyper-parameters are: $\alpha_g = 0.0001$, $\alpha_d = 0.0001$, $n_{dis} = 5$. FID is 7.40.

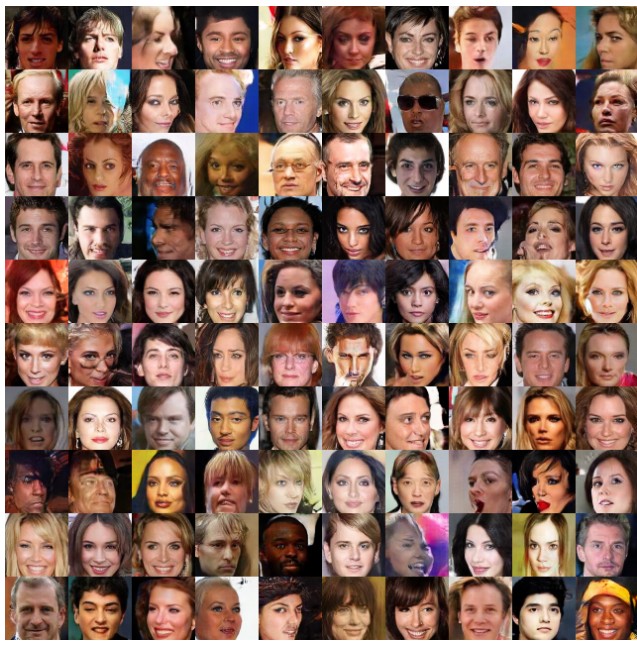

Figure 62: Generated samples from the best run of SN (diff. $\gamma$) in CelebA. The hyper-parameters are: $\alpha_g = 0.0001$, $\alpha_d = 0.0001$, $n_{dis} = 5$. FID is 7.29.

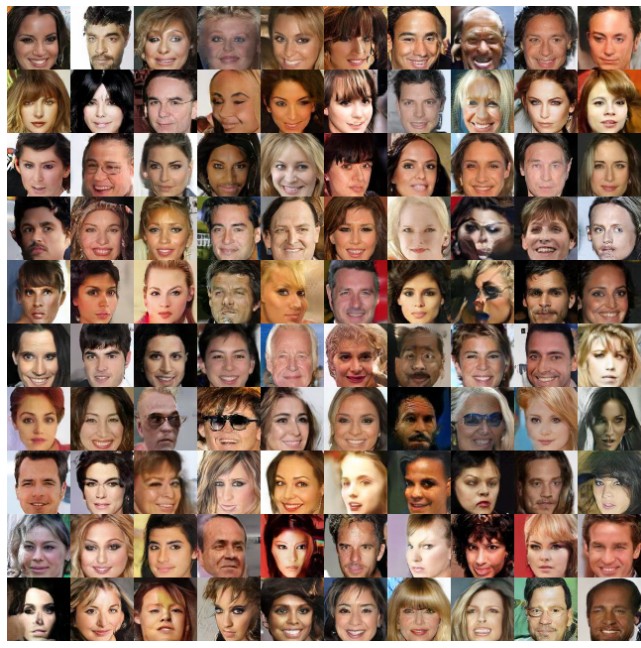

Figure 63: Generated samples from the best run of SN in CelebA. The hyper-parameters are: $\alpha_g = 0.0002$, $\alpha_d = 0.0001$, $n_{dis} = 1$. FID is 8.34.

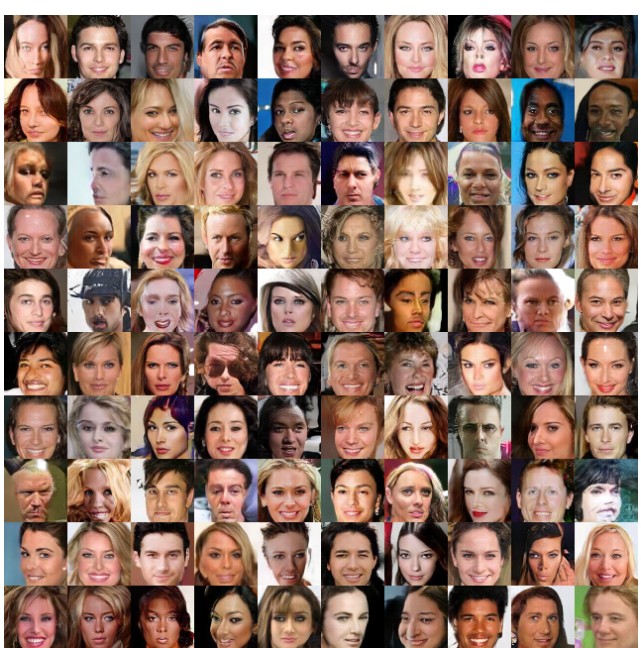

Figure 64: Generated samples from the best run of BSN in CelebA. The hyper-parameters are: $\alpha_g = 0.0002$, $\alpha_d = 0.0001$, $n_{dis} = 1$. FID is 8.06.

| |
|---|
| $z \in \mathbb{R}^{128} \sim \mathcal{N}(0, I)$ |
| Fully connected ($4 \times 4 \times 1024$). |
| ResNet-up ($c = 1024$). |
| ResNet-up ($c = 512$). |
| ResNet-up ($c = 256$). |
| ResNet-up ($c = 128$). |
| ResNet-up ($c = 64$). |
| BN. ReLU. Convolution ($c = 3, k = 3, s = 1$). Tanh |

Table 8: Generator network architectures for ILSVRC2012 experiments (from [30]). BN stands for batch normalization. $c$ stands for number of channels. $k$ stands for kernel size. $s$ stands for stride.

| |
|---|
| **Direct connection** |
| BN. ReLU. Unpooling(2). Convolution ($k = 3, s = 1$). |
| BN. ReLU. Convolution ($k = 3, s = 1$). |
| **Shortcut connection** |
| Unpooling(2). Convolution ($k = 1, s = 1$). |

Table 9: ResNet-up network architectures for ILSVRC2012 experiments (from [30]). BN stands for batch normalization. $k$ stands for kernel size. $s$ stands for stride.

# S   Experimental Details and Additional Results on ILSVRC2012

## S.1   Experimental Details

The network architectures are shown in Tables 8 to 13. All experiments are run for 500k iterations. Discriminator batch size is 16. Generator batch size is 32. The optimizer is Adam. $\alpha_g = 0.002, \alpha_d = 0.002, \beta_1 = 0.0, \beta_2 = 0.9, n_{dis} = 5$ We use hinge loss with the popular SN implementation [30].

## S.2   Training Curves

Figs. 65 and 66 show the inception score and FID of SN and BSN during training.

For SN, we can see that the runs with scale=1.0/1.2/1.4 have similar performance throughout training. When scale=1.6, the performance is much worse.

For BSN, the runs with scale=1.2/1.4 perform better than SN runs throughout the training. When scale=1.6, BSN has similar performance as SN at the early stage of training, and is slightly better at the end. When scale=1.0, the performance is very bad as there is gradient vanishing problem.

## S.3   Generated Images

Figs. 67 to 74 show the generated images from the run with the best inception score for SN and BSN with different scale parameters.

| |
|---|
| $x \in \mathbb{R}^{128 \times 128 \times 3}$ |
| ResNet-first ($c = 64$). |
| ResNet-down ($c = 128$). |
| ResNet-down ($c = 256$). |
| ResNet-down ($c = 512$). |
| ResNet-down ($c = 1024$). |
| ResNet ($c = 1024$). |
| ReLU. Global pooling. Fully connected (1). |

Table 10: Discriminator network architectures for ILSVRC2012 experiments (from [30]). BN stands for batch normalization. $c$ stands for number of channels. $k$ stands for kernel size. $s$ stands for stride.

| Direct connection |
| --- |
| ReLU. Convolution ($k = 3, s = 1$). |
| ReLU. Convolution ($k = 3, s = 1$). Average pooling(2). |
| **Shortcut connection** |
| Convolution ($k = 1, s = 1$). Average pooling(2). |

Table 11: ResNet-down network architectures for ILSVRC2012 experiments (from [30]). $k$ stands for kernel size. $s$ stands for stride.

| Direct connection |
| --- |
| Convolution ($k = 3, s = 1$). |
| ReLU. Convolution ($k = 3, s = 1$). Average pooling(2). |
| **Shortcut connection** |
| Average pooling(2). Convolution ($k = 1, s = 1$). |

Table 12: ResNet-first network architectures for ILSVRC2012 experiments (from [30]). $k$ stands for kernel size. $s$ stands for stride.

| Direct connection |
| --- |
| ReLU. Convolution ($k = 3, s = 1$). |
| ReLU. Convolution ($k = 3, s = 1$). |
| **Shortcut connection** |
| Convolution ($k = 1, s = 1$). |

Table 13: ResNet network architectures for ILSVRC2012 experiments (from [30]). $k$ stands for kernel size. $s$ stands for stride.

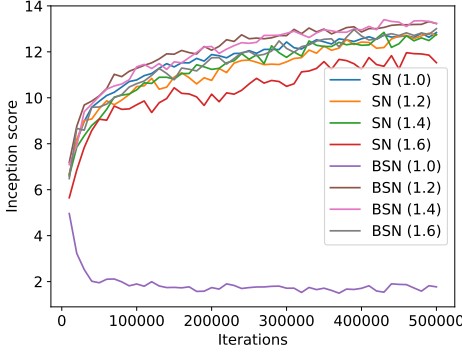

Figure 65: Inception score in ILSVRC2012. The results are averaged over 5 random seeds.

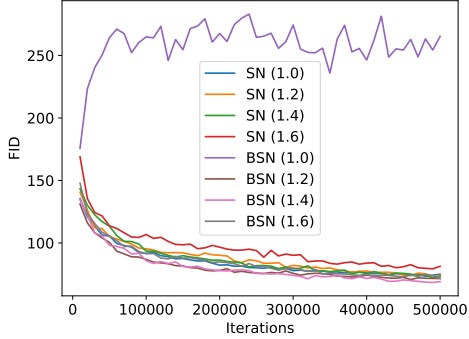

Figure 66: FID in ILSVRC2012. The results are averaged over 5 random seeds.

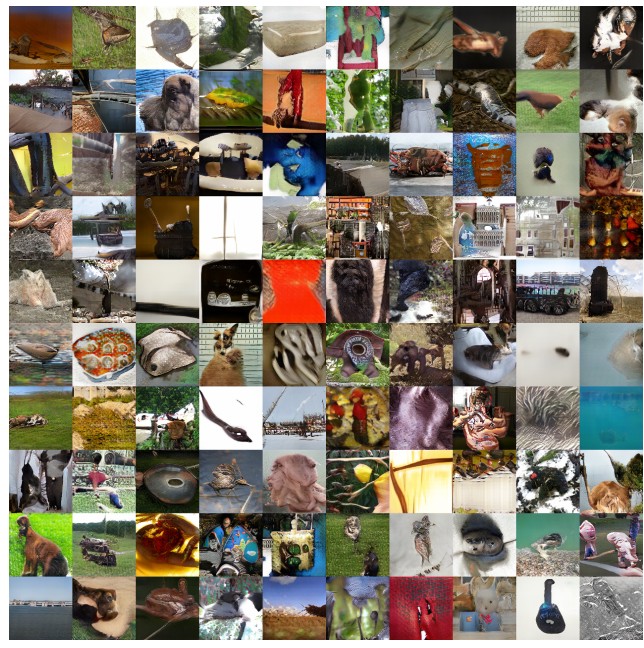

Figure 67: Generated samples from the best run of SN (scale=1.0) in ILSVRC2012. Inception score is 13.50. FID is 72.18.

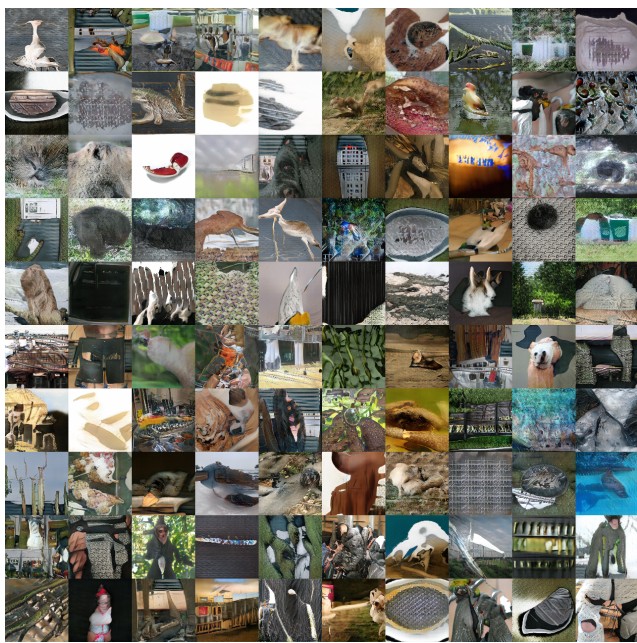

Figure 68: Generated samples from the best run of SN (scale=1.2) in ILSVRC2012. Inception score is 13.04. FID is 72.51.

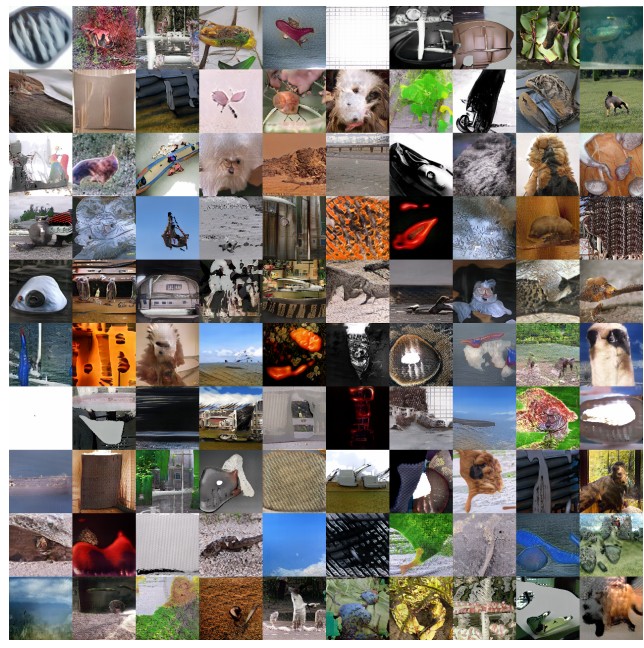

Figure 69: Generated samples from the best run of SN (scale=1.4) in ILSVRC2012. Inception score is 13.04. FID is 69.12.

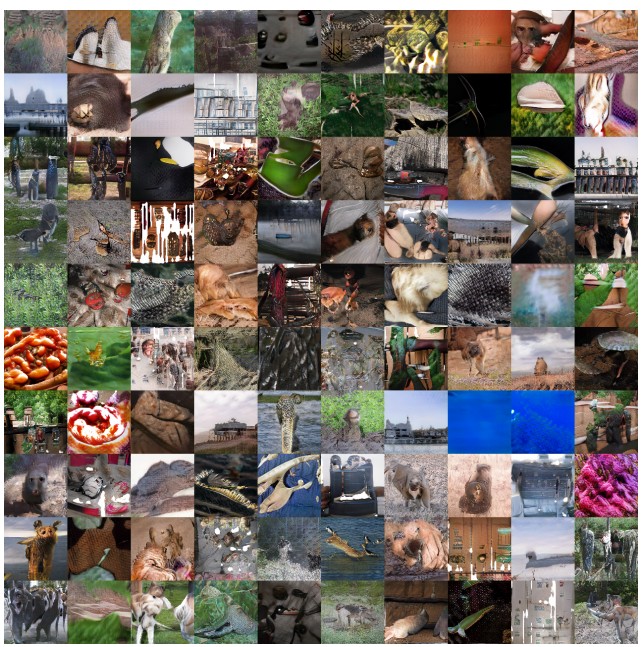

Figure 70: Generated samples from the best run of SN (scale=1.6) in ILSVRC2012. Inception score is 12.62. FID is 70.36.

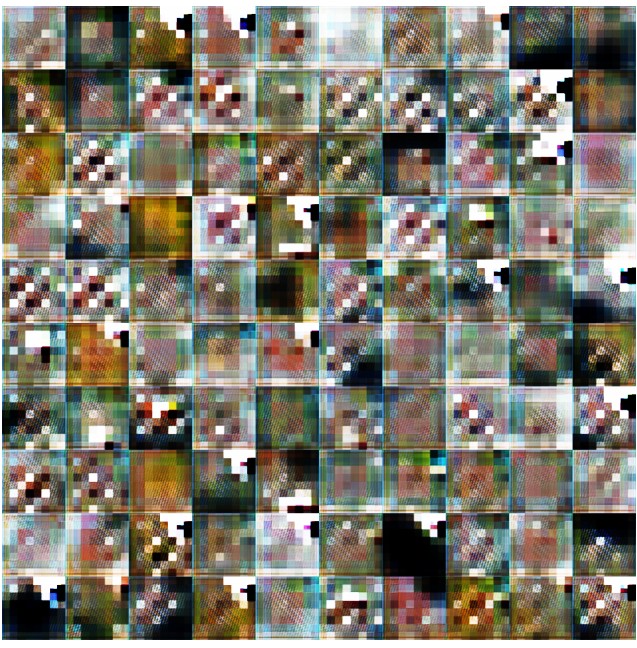

Figure 71: Generated samples from the best run of BSN (scale=1.0) in ILSVRC2012. Inception score is 2.07. FID is 242.51.

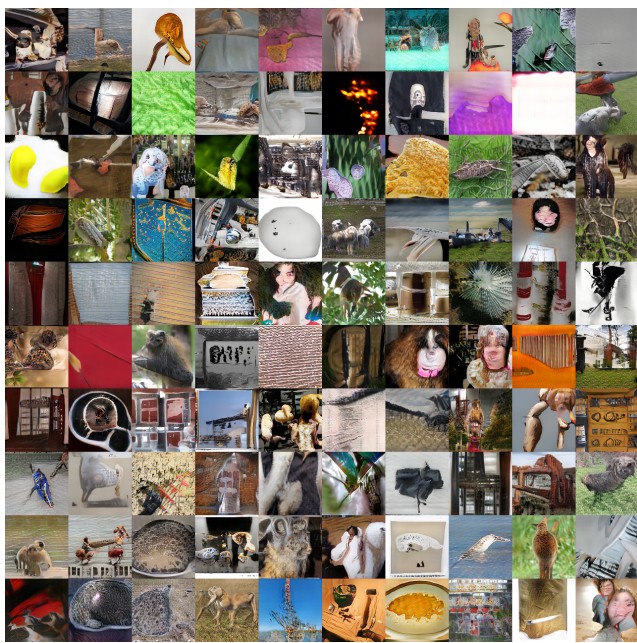

Figure 72: Generated samples from the best run of BSN (scale=1.2) in ILSVRC2012. Inception score is 13.55. FID is 71.30.

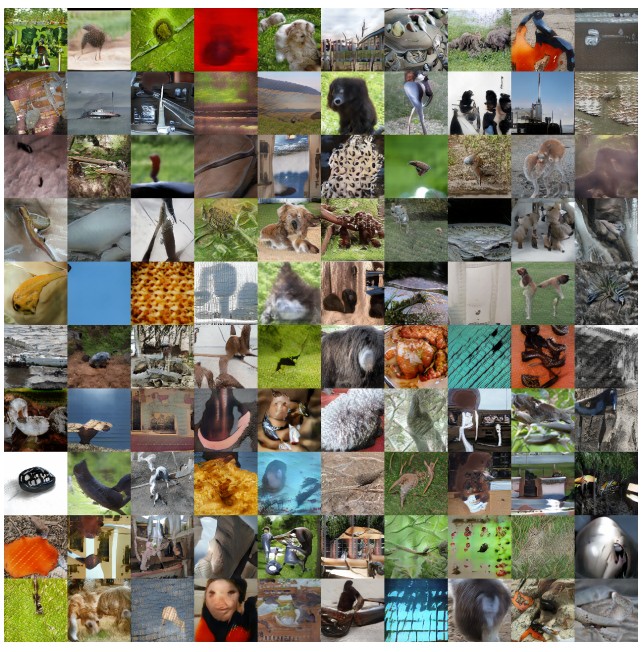

Figure 73: Generated samples from the best run of BSN (scale=1.4) in ILSVRC2012. Inception score is 13.63. FID is 70.88.

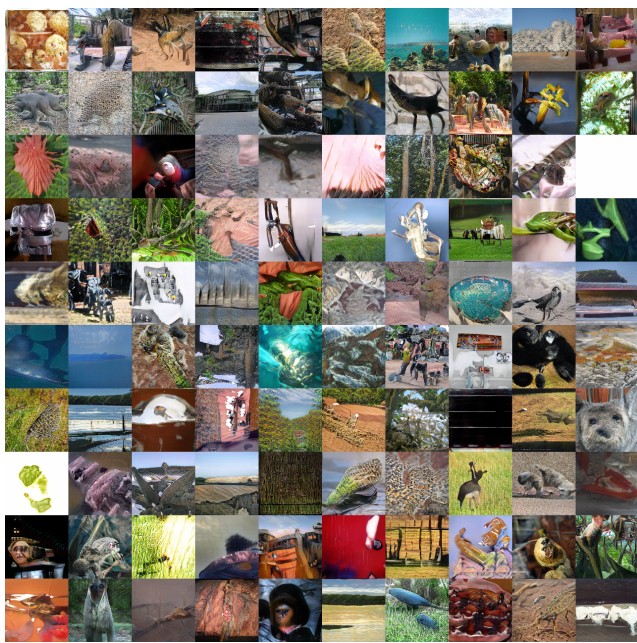

Figure 74: Generated samples from the best run of BSN (scale=1.6) in ILSVRC2012. Inception score is 13.24. FID is 69.06.

# T   Details on Computation Resources

All the experiments were run on a public cluster: Bridges-2 system at the Pittsburgh Supercomputing Center (PSC) with NVIDIA Tesla V100 GPUs. All the experiments took around 30k GPU hours.