# OpenReview forum: "Why Spectral Normalization Stabilizes GANs: Analysis and Improvements"
_NeurIPS.cc/2021/Conference — NeurIPS 2021 Poster_

### Official Review · Reviewer_8Yem · 2021-07-09

**Rating:** 6
**Confidence:** 4

**Summary:**

The paper analyzes the benefits of the spectral norm on GAN: preventing gradient exploding and gradient vanishing. They prove that SN can control the variance of weights, which can prevent gradient vanishing, following the logic of LeCun initialization. They further propose a bidirectional SN, which is an improvement based on SN, following the logic of Xariver initialization. They conduct experiments on multiple datasets with DCGAN structure.

**Limitations And Societal Impact:**

The paper mentions the limits and leaves them in future work.

**Main Review:**

Originality (+): To my best knowledge, this work is the first one that connects SN with LeCun initialization and formally proves that SN can control the variance of weights and prevent gradient vanishing. But preventing gradient explosion is relatively well-known and less surprising. Overall, it can provide people a more theoretical understanding of SN.

Quality:

Pros:

1. I think the proof itself is sound, but I have some questions about Theorem 2 (see cons).

2. Plots are clear and informative.

3. I think it's a complete piece of work.

Cons:

1. Theorem 2 doesn't provide a direct bound or analysis on the gradient, while Theorem 1 does. Instead, they analyze the variance of weights and justify the gradient vanishing in a roundabout way.

2. In Theorem 2, there is a gap between assumption and real GAN training: LeCun init assumes the activation layers are symmetrical, while GAN usually uses LeakReLU, which lowers the variance of the input and then weakens the story.

3. The lower bound in Theorem 2 could be far smaller than 1: 1/(16K^2 * max{m,n} * log(min{m,n})  (line 636). This means the variance of activation will get smaller and smaller. This situation will get worse with the effect of unsymmetrical activation layers. So from this bound, I cannot see the variance gets preserved, and the training can get saturated actually.

4. From Fig. 4, the gradient norm does get preserved. So I concern if Theorem 2 is the correct way to prove this.

Clarity: The submission is clearly written and easy to follow.

Significance:

Pros:

1. I think the message that SN can prevent gradient explosion and vanishing is significant and the message itself should be correct by a large chance.

2. The proposed BSN and BSSN can improve the results of SN.

Cons: Besides the concerns mentioned above, the experiments have some flaws that may make researchers hesitate to use the proposed methods in their work.

1. They only show the success with DCGAN structure, but not with more complex structures (e.g. ResNet and BigGAN) for the majority of the experiments. In the only experiment with the ResNet structure, BSN doesn't show the advantage.

2. Tunning the weight scaling seems too cumbersome (i.e. pick the best from [0.7, 0.8, 0.9, 1.0, 1.2, 1.4, 1.6])

I will raise my score if concerns get addressed.

**Time Spent Reviewing:**

6

---

> ### Author Response · Authors · 2021-08-10
> **Rebuttal**
>
> Thank you very much for your insightful questions and suggestions! In the following, we repeat each of your questions (marked with ‘Q’), followed by our response (marked with ‘A’).
>
> ---
>
> Q: Theorem 2 doesn't provide a direct bound or analysis on the gradient, while Theorem 1 does. Instead, they analyze the variance of weights and justify the gradient vanishing in a roundabout way.
>
> [ * ] A: The connection between weight variances and gradient scales is studied in LeCun et al. (1998), which we cited and briefly discussed in Section 4 from line 208 to 220. Due to the space constraint, we did not repeat their mathematical details. We explain it here, and will add this discussion to our paper. Assume that the inputs x_i (i=1,...,n) to a layer in the neural network are uncorrelated with variance 1, where n is the fan-in of the layer. If we ensure that the weights in the layer are zero-mean with variance 1/n, then for any output unit y_j in the layer, we have Var(y_j)=\sum_{i=1}^n w_{ji}^2=1, equal to that of the inputs. This means if we ensure that the variance of parameters is 1/n, the variance of the internal outputs of the neural network remains the same, no matter how many layers there are. However, (1) if the parameter variances are larger than 1/n, the scale (variance) of the internal outputs are getting larger in later layers, and could finally get saturated by non-linear activation functions or loss functions (e.g., sigmoid, tanh), which causes small gradients; (2) if the parameter variances are small, the gradient will also be small, as gradients are scaled by the parameters (Appendix A). We agree that this is more roundabout than Proposition 1, but we believe it illustrates how SN mitigates gradient vanishing. We will add these explanations to the paper to make it more self-contained.
>
> ---
>
> Q: In Theorem 2, there is a gap between assumption and real GAN training: LeCun init assumes the activation layers are symmetrical, while GAN usually uses LeakReLU, which lowers the variance of the input and then weakens the story.
>
> A: Just to clarify, Theorem 2 does not assume symmetrical activation layers; it is a statement about a spectrally-normalized weight matrix (independent of activation). To connect Theorem 2 to the gradient vanishing problem, we initially used the LeCun initialization [25] in Section 4, which as you correctly pointed out, assumes symmetric activation layers. However, one can (and we do, in Section 5) connect the two using the analysis of other initialization papers (e.g., Kaiming initialization [17]) that do *not* assume symmetric activation. Indeed, this is the motivation behind the scaling technique in Section 5.2, which draws inspiration from Kaiming initialization, which considers leaky ReLU activations. We will make this clearer in the paper.
>
> ---
>
> Q: The lower bound in Theorem 2 could be far smaller than 1: 1/(16K^2 * max{m,n} * log(min{m,n}) (line 636). This means the variance of activation will get smaller and smaller. This situation will get worse with the effect of unsymmetrical activation layers. So from this bound, I cannot see the variance gets preserved, and the training can get saturated actually. From Fig. 4, the gradient norm does get preserved. So I concern if Theorem 2 is the correct way to prove this.
>
> A: This is a great question. For some models (e.g., convolutional networks with small kernels), the lower bound is fairly close to the upper bound, but we agree that the lower bound may be too loose to be significant for larger models with high fan-in and/or fan-out. That being said, we observe that empirically, the actual variances seem to track the *upper bound* proved in Theorem 2, rather than the lower bound (Figure 7). We are not currently sure how to explain this, including whether it is due to looseness in our lower bound analysis or if it is due to something unrelated to spectral normalization. We believe that answering this question would be an interesting direction for future work.
>
> ---
>
> Q: They only show the success with DCGAN structure, but not with more complex structures (e.g. ResNet and BigGAN) for the majority of the experiments. In the only experiment with the ResNet structure, BSN doesn't show the advantage.
>
> A: Our final proposed approach is actually not BSN but BSSN (which combines BSN and SSN). We observe in the paper that BSN alone is not sufficient to get good performance always; combining it with scaling is the right thing to do. Indeed, BSSN does improve the performance of ResNets on ILSVRC2021 in terms of both inception score and FID (Table 1). (Also, we were limited in terms of the number of experiments we could run on these larger models due to financial and computational constraints.)
>
> ---
>
> Q: Tunning the weight scaling seems too cumbersome (i.e. pick the best from [0.7, 0.8, 0.9, 1.0, 1.2, 1.4, 1.6])
>
> A: We ran a wide range of weight scaling as a sensitivity study for showing readers how weight scaling affects the performance. However, we found that the range of the optimal scales across all four datasets is actually pretty small (within 1.0 ~ 1.4). Therefore, we would expect that tuning this parameter does not require as much work as our experiments might suggest. Making this process automatic (e.g., making this parameter learnable) could be an interesting future work.

---

> > ### Comment · Reviewer_8Yem · 2021-08-24
> > **further question**
> >
> > I thank the authors' replies, and they address most of my questions.
> >
> > I have a further question on question 2 (Q: In Theorem 2, there is a gap between assumption and real GAN training: LeCun init assumes the activation layers are symmetrical, while GAN usually uses LeakReLU, which lowers the variance of the input and then weakens the story.)
> >
> > I agree with the authors that Kaiming init is applicable to unsymmetrical activations, but it uses a numerical larger than 1 (e.g. sqrt 2) to compensate for the reducing variance brought by unsymmetrical activations. Just like we use gain when doing initialization in Pytorch code. But in theorem 2, the upper bound's numerator is 1, and the discriminator uses unsymmetrical activations, and the orange line in Fig. 6 shows SN even prefers a scalar lower than 1. All these three factors make me wonder how SN keeps the variance.
> >
> > Thanks,
> >
> > Reviewer 8Yem

---

> > > ### Author Response · Authors · 2021-08-26
> > > **Response**
> > >
> > > Q: I have a further question on question 2 (Q: In Theorem 2, there is a gap between assumption and real GAN training: LeCun init assumes the activation layers are symmetrical, while GAN usually uses LeakReLU, which lowers the variance of the input and then weakens the story.)
> > > I agree with the authors that Kaiming init is applicable to unsymmetrical activations, but it uses a numerical larger than 1 (e.g. sqrt 2) to compensate for the reducing variance brought by unsymmetrical activations. Just like we use gain when doing initialization in Pytorch code. But in theorem 2, the upper bound's numerator is 1, and the discriminator uses unsymmetrical activations, and the orange line in Fig. 6 shows SN even prefers a scalar lower than 1. All these three factors make me wonder how SN keeps the variance.
> > >
> > > A: This is (another) great question. In fact, even for LeCun initialization, we observed that there are two good modes for the scaling factor: one is larger than 1, as predicted by Kaiming initialization; the other is smaller than 1, which cannot be explained by current initialization theories. This is illustrated in Figure 8. Similar to this, we observed that for spectral normalization, there are also two good modes of scaling. (The orange curve in Figure 6 only shows one mode. The extended version of it is the orange curve in Figure 8, which gives a complete view of the two modes.) We showed in Appendix M that the two modes in initialization and the two modes in spectral normalization have an interesting correspondence, which may back up our insights on the relationship between initialization techniques and spectral normalization.
> > >
> > > Given these empirical observations, we believe that to fully answer your question, we need to first understand why LeCun initialization has that good mode of scaling smaller than 1, which cannot be explained by current initialization literature. This is a very interesting question, but we do not currently have a full understanding and would like to study it in future work.

---

> > > > ### Comment · Reviewer_8Yem · 2021-08-28
> > > > **thanks**
> > > >
> > > > I thank the authors' replies. Though I am still not sure if controlling the variance as Lecun init is the key for SN's success, I believe this work does provide an explanation to the committee and will encourage researchers to learn SN more from the theoretical perspective.
> > > >
> > > > I decide to raise my score accordingly.

---

### Official Review · Reviewer_NDw4 · 2021-07-11

**Rating:** 8
**Confidence:** 4

**Summary:**

This paper provides insights towards understanding the stability of spectral normalization GANs (SNGANs), and proposes a new framework (BSSN) based on the proposed theoretical insights.

**Limitations And Societal Impact:**

Perhaps the authors could improve the paper by answering/clarifying questions 1) and 2) mentioned in the main review.

**Main Review:**

Overall this is a well-written paper, it provides two theoretical insights on the stability of SNGAN: SN prevents exploding (in Sec. 3) and vanishing (in Sec. 4) gradients. Based on the theoretical analysis, this paper also proposes the BSSN framework that incorporates the theoretical insights. See detailed comments/questions below:

1) In Theorem 1, how often does the constraint of $\Pi_{i=1}^Lc_i=1$ hold in practice? Perhaps the author can provide some examples/justifications of this assumption?

2) Comparing Theorem 2 and Theorem 3, it seems that the author believe that the BSSN is better than the SN because the BSSN can controls the variance at the scale of $\frac{2}{(n_i+m_i)}$ while SN controls the variance at the scale of $\frac{1}{\max\{n_i,m_i\}}$, is there any theoretical justification that $\frac{2}{(n_i+m_i)}$ is better than $\frac{1}{\max\{n_i,m_i\}}$?

3) The reviewer appreciates the way that the authors present their theoretical results: each proposed theorem comes along with experiments that justifying their observations (Fig. 2-3 for Thm. 1 in Section 3; Fig. 4-6 for Thm. 2 in Section 4; and Fig. 7 for Thm. 3 in Section 5).

4) The reviewer is not an expert in GANs experiments, but the reviewer finds the proposed BSSN framework indeed performs better under both FID and IS, and the numerical experiments seem reasonable and promising.

**Time Spent Reviewing:**

5 hours

---

> ### Author Response · Authors · 2021-08-10
> **Rebuttal**
>
> Thank you very much for your great questions! In the following, we repeat each of your questions (marked with ‘Q’), followed by our response (marked with ‘A’).
>
> ---
>
> Q: In Theorem 1, how often does the constraint of $\prod_{i=1}^L{c_i}= 1$ hold in practice? Perhaps the author can provide some examples/justifications of this assumption?
>
> A: To clarify, we did not mean that during training $\prod_{i=1}^L{c_i}= 1$ necessarily holds. The message of Theorem 1 is that for any set of parameters $\{w_1,...,w_L\}$, the scaling in the equation between line 179 and 180 always minimizes the gradient’s Frobenius norm without changing the network’s behavior on the input samples. The constraint $\prod_{i=1}^L{c_i}= 1$ is just a shorthand for saying ‘without changing the network’s behavior on the input samples’, which holds thanks to Proposition 2. We will clarify this in the revision.
>
> ---
>
> Q: Comparing Theorem 2 and Theorem 3, it seems that the author believe that the BSSN is better than the SN because the BSSN can controls the variance at the scale of $2/(n_i+m_i)$ while SN controls the variance at the scale of 1/max{n_i,m_i}, is there any theoretical justification that $2/(n_i+m_i)$  is better than 1/max{n_i,m_i}
>
> A: Your understanding is correct: the benefit of BSN over SN is from the different scales of the variances. The variance of SN mimics LeCun initialization (Section 4), and the variance of BSN mimics Xavier initialization (Section 5). We believe the benefits of BSN over SN can be explained by the same reasoning that explains the benefit of Xavier initialization over LeCun initialization, which are discussed in detail in Glorot & Bengio (2010), and cited in line 274. We recap their results here, and will add a more detailed explanation in our paper:
>
> Glorot and Bengio found that controlling the variance of internal outputs in the feed-forward phase is not sufficient to avoid vanishing gradient (see the first answer to Reviewer 8Yem, marked with [ * ]); gradients can still vanish because of the computation during back-propagation. To ensure the stability of the back-propagation phase, we need to set the variance of weights to $1 / m$, where $m$ is fan-out of the layer (the reasoning and derivation of it are the same as [ * ], but changing inputs to back-propagated gradients). To balance the constraints on the feed-forward outputs and back-propagated gradients, they choose to set the parameter variances as $2/(n+m)$. We will add these explanations from the prior work to the paper to make it more self-contained.

---

### Official Review · Reviewer_zyay · 2021-07-15

**Rating:** 7
**Confidence:** 3

**Summary:**

The paper offers a more insightful analysis on Spectral Normalization (SN) and shows that SN controls two important failure modes of GAN: exploding and vanishing gradients. Specifically, the paper provides the theories to prove that SN upper-bounds the gradients and controls the variance of weights closely to LeCun initialization. Furthermore, given the new understanding of the connection between SN and LeCun initialization, the paper proposes a simple modification of SN to control both the variances of internal outputs and the variance of backpropagation gradients, which mimics the way of Xavier initialization. It also introduces new scaling weights, motivated by Kaming initialization, to obtain better performance in practice. The paper provides ablation studies and experiments on multiple benchmark datasets.

**Main Review:**

Pros
The paper is well-written, and contributions are verified with empirical experiments on benchmark datasets.  Understanding GAN is important, and this paper offers a useful contribution.

Cons
The analysis of gradient exploding and gradient vanishing have experimented on two different datasets: MNIST and CIFAR-10. Do both issues happen on the same dataset? It is good to verify these problems on other datasets as well.

BSN failed on the ILSVRC2012 dataset, do the authors have any insights why?

What GAN objective function is used in the implementation? In my experience, SN (Miyato et al.,) does work only well with hinge loss objectives. Do authors have any insights into that from the analysis? Also, does the improved SN proposed work well with other loss functions?


**Time Spent Reviewing:**

Four hours

---

> ### Author Response · Authors · 2021-08-10
> **Rebuttal**
>
> Thank you very much for your insightful comments! In the following, we repeat each of your questions (marked with ‘Q’), followed by our response (marked with ‘A’).
>
> ---
>
> Q: The analysis of gradient exploding and gradient vanishing have experimented on two different datasets: MNIST and CIFAR-10. Do both issues happen on the same dataset? It is good to verify these problems on other datasets as well.
>
> A: It has been widely-documented in prior work that gradient exploding and vanishing causes GAN training to fail on different datasets and architectures (e.g., [2,8]). Because of this, we only reproduced the phenomenon on two datasets for illustration purposes. We will run it on other datasets and append the results to the paper.
>
> ---
>
> Q: BSN failed on the ILSVRC2012 dataset, do the authors have any insights why?
>
> A: As explained in Section 5.2, the inherent scaling in BSN (i.e., Xavier initialization) is not always the best. Here, by “inherent scaling”, we mean the approach without any additional or special scaling factor. Note that even on other datasets, SN’s and BSN’s inherent scalings are also not the best (Table 1). The networks used in ILSVRC2012 experiments have more layers than other datasets, and therefore, the scaling error gets amplified more and causes more significant performance degradation. Note that this is exactly why we proposed weight scaling in Section 5.2, to mitigate this issue.
>
> ---
>
> Q: What GAN objective function is used in the implementation? In my experience, SN (Miyato et al.,) does work only well with hinge loss objectives. Do authors have any insights into that from the analysis? Also, does the improved SN proposed work well with other loss functions?
>
> A: We used hinge loss, following the SN paper (Miyato et al.).  We did not try SN or BSSN on other loss functions in this paper, but we did see that SN has been successfully applied with other losses as well [28,21,27]. Your proposal of studying the interaction between SN (or BSSN) and loss functions is very interesting! But probably we will need a different approach, as we do not think our analysis approach can be directly used to study this problem. This could be an interesting future work direction.

---

### Official Review · Reviewer_oz6n · 2021-07-16

**Rating:** 7
**Confidence:** 4

**Summary:**

This submission analyzes spectral normalization (SN), an widely used and effective method for improving GANs, from the viewpoint of its effect on the gradients of the parameters of the networks (as opposed to the gradients with regards to their inputs).
Doing so, it shows that SN controls for vanishing and exploding gradients in the discriminator. It also provides a possible explanation as to why the usual implementation of SN is more effective than its "formally correct" counterpart.
Finally, drawing links with the popular LeCun initialization, it proposes an improved variant of Spectral Normalization.





**Limitations And Societal Impact:**

Limitations are openly discussed.
Potential societal impacts are adequately adressed.

**Main Review:**

Originality:
Analyzing gradients is a natural and fairly well-studied way to consider spectral normalization. However, by specifically studying the weights of the discriminator, this work is able to find novel, surprising, results and applications, including links with initialization methods. While the related work paragraph is succinct, much of the actual related work is discussed thoroughly in the relevant sections of the paper.

Quality:
The theoretical analysis seems of good quality. Experiments are comprehensive on multiple relevant datasets. The modifications for SN proposed in section 5 are less convincing. While theoretically strong, they seem to only result in a marginal (and not always consistent) improvement in practice.

Clarity:
The submission is clearly written and well organized.

Significance:
The paper provides some new understanding of Spectral Normalization, a crucial component of GANs. The findings are interesting theoretically and could lead to future improvements. Importantly, the paper opens a new direction for future work.

**Time Spent Reviewing:**

3

---

> ### Author Response · Authors · 2021-08-10
> **Rebuttal**
>
> Thank you very much for your feedback on our paper!

---

### Author Response · Authors · 2021-08-10
**Rebuttal**

We would like to thank all reviewers for the high-quality reviews and constructive comments! We did our best in answering all your questions, and we'll be happy to answer any additional questions during the discussion period. Thank you again for the time reviewing our work!

---

### Decision · Program_Chairs · 2021-09-27

**Decision:**

Accept (Poster)

**Comment:**

The submission discusses the benefits of spectral normalization for GAN training. Initial reviewer assessment was mostly positive while one reviewer was a bit more concerned. After a discussion with the authors the reviewer appreciated the contributions while remaining unconvinced that controlling the variance via Lecun initialization is the key for SN's success. AC thinks this paper provides interesting insights that could spur future research.